# Application of optical tweezer technology reveals that PfEBA and PfRH ligands, not PfMSP1, play a central role in *Plasmodium falciparum* merozoite-erythrocyte attachment

**Emma Kals**[1,2], **Morten Kals**[2], **Rebecca A. Lees**[3,4], **Viola Introini**[1,2,5], **Alison Kemp**[1], **Eleanor Silvester**[1], **Christine R. Collins**[4], **Trishant Umrekar**[3,4], **Jurij Kotar**[2], **Pietro Cicuta**[2], **Julian C. Rayner**[1] *

1 Cambridge Institute for Medical Research, University of Cambridge, Cambridge, United Kingdom, 2 Cavendish Laboratory, University of Cambridge, Cambridge, United Kingdom, 3 Institute of Structural and Molecular Biology, Department of Biological Sciences, Birkbeck, University of London, London, United Kingdom, 4 Malaria Biochemistry Laboratory, The Francis Crick Institute, London, United Kingdom, 5 EMBL Barcelona, Barcelona, Spain

* jcr1003@cam.ac.uk

## Abstract

Malaria pathogenesis and parasite multiplication depend on the ability of *Plasmodium* merozoites to invade human erythrocytes. Invasion is a complex multi-step process involving multiple parasite proteins which can differ between species and has been most extensively studied in *P. falciparum*. However, dissecting the precise role of individual proteins has to date been limited by the availability of quantifiable phenotypic assays. In this study, we apply a new approach to assigning function to invasion proteins by using optical tweezers to directly manipulate recently egressed *P. falciparum* merozoites and erythrocytes and quantify the strength of attachment between them, as well as the frequency with which such attachments occur. Using a range of inhibitors, antibodies, and genetically modified strains including some generated specifically for this work, we quantitated the contribution of individual *P. falciparum* proteins to these merozoite-erythrocyte attachment interactions. Conditional deletion of the major *P. falciparum* merozoite surface protein PfMSP1, long thought to play a central role in initial attachment, had no impact on the force needed to pull merozoites and erythrocytes apart, whereas interventions that disrupted the function of several members of the EBA-175 like Antigen (PfEBA) family and Reticulocyte Binding Protein Homologue (PfRH) invasion ligand families did have a significant negative impact on attachment. Deletion of individual PfEBA and PfRH ligands reinforced the known redundancy within these families, with the deletion of some ligands impacting detachment force while others did not. By comparing over 4000 individual merozoite-erythrocyte interactions in a range of conditions and strains, we establish that the PfEBA/PfRH families play a central role in *P. falciparum* merozoite attachment, not the major merozoite surface protein PfMSP1.

**Data Availability Statement:** The Python code and associated data files that support the findings of this study are deposited at https://zenodo.org/doi/10.5281/zenodo.10645167. Summaries of relevant data are included in the manuscript and its supporting information files. Protocols can be found at: DOI: dx.doi.org/10.17504/protocols.io.5qpvo3zmzv4o/v1 and DOI: dx.doi.org/10.17504/protocols.io.q26g7peo9gwz/v1.

**Funding:** This work was funded by Wellcome (https://wellcome.org/), grant numbers (220266/Z/20/Z to JR, AK, ES) and (222323/Z/21/Z to EK); EU-EC Marie Curie ITN PyMot (https://marie-sklodowska-curie-actions.ec.europa.eu/) (955910 to MK) and Engineering and Physical Sciences Research Council (EPSRC) grant (https://www.ukri.org/councils/epsrc/) (EP/W004453/1 to PC and JK). The funders played no role in the study design, data collection and analysis, decision to publish, or preparation of the manuscript.

**Competing interests:** The authors have declared that no competing interests exist.

## Author summary

Malaria is a devastating disease caused by a parasitic infection. The deadliest species is *Plasmodium falciparum*, which causes more than 600,000 deaths annually. The *Plasmodium* life cycle is complex, but all the symptoms of malaria are caused when the parasites replicate in human red blood cells. Replication depends on the invasion of the red blood cells by the parasites, a process involving multiple molecular interactions and multiple steps. Invasion begins with the attachment of the parasite to the red blood cell, making this step of particular interest in the development of new therapeutics. We used an optical tweezer assay to directly measure the binding force between individual parasites and red blood cells, and combined this assay with a range of molecular and genetic tools that target specific interactions known to have a role in invasion. This approach showed that loss of a protein commonly thought to be critical to the early stages of invasion, PfMSP1, had no effect on attachment strength, whereas disruptions of several members from two families of proteins (the Erythrocyte Binding Like protein family and the Reticulocyte Binding-like protein family) did affect attachment strength.

## Introduction

*Plasmodium falciparum* parasites cause >600,000 malaria deaths annually [1]. The *P. falciparum* lifecycle is complex, but all the clinical symptoms are caused by the blood stage of the infection [2]. This blood stage begins when extracellular forms called merozoites egress from the liver, after which they rapidly reinvade new erythrocytes and then develop within the erythrocyte over the next 48 hours, resulting in the production of 26–36 new merozoites that egress to begin the cycle again [3–7]. Invasion is, therefore, essential for parasite multiplication and as merozoites are exposed to the antibody-mediated immune system, invasion has long been a focus of vaccine development [8]. Despite the speed with which it occurs, invasion is a complex process involving multiple *Plasmodium* proteins and involves both specific and non-specific receptor-ligand interactions between merozoite and erythrocyte. Invasion begins with attachment of the parasite to the erythrocyte, followed by reorientation to position the parasite to invade, then strong membrane wrapping of the erythrocyte membrane around the merozoite occurs until finally a tight junction between the two cells is formed which is actively passed around the merozoite, internalising the parasite within a parasitophorous vacuole [3–5,9].

Extensive work over multiple decades has identified many *P. falciparum* proteins involved in invasion, and in some cases assigned them to specific steps. The initial weak attachment of the merozoite to the erythrocyte has been ascribed to multiple Merozoite Surface Proteins (PfMSPs), particularly Merozoite Surface Protein 1 (PfMSP1) [10–12]. Deformation of the erythrocyte then increases and the merozoite reorientates itself so its apex (the wider end) points at the erythrocyte [13,14]. This phase is thought to involve the Erythrocyte Binding Antigen (PfEBA) family and Reticulocyte Binding Protein Homologue (PfRH) family of proteins which interact with a range of erythrocyte receptors both known and unknown [15]. The binding of different PfEBA and PfRH members is considered redundant and overlapping, with different *P. falciparum* strains relying on different family members. Deformation then relaxes and a distinct member of the PfRH family PfRH5, the only essential member of the family [16,17], binds Basigin in the erythrocyte membrane as part of a complex with PfPTRAMP–PfCSS–PfRipr–PfCyRPA–PfRh5 (PCRCR) [18,19]. Next the tight junction forms through the interaction of PfAMA1 and PfRON2 [20] and an actin-myosin motor then pulls the tight junction around the merozoite until the merozoite enters the erythrocyte surrounded

by the parasitophorous vacuole [21–25]. Behind the tight junction, multiple proteases, including subtilisin-like serine protease 2 (PfSUB2), cleave merozoite surface proteins, including PfRH and PfEBA family members and PfAMA1 [26–28]. Invasion ends with the sealing of the parasitophorous vacuole and erythrocyte membrane, which is catalysed by PfSUB2 [27]. The processes, proteins and ligand-receptor interactions that take place during invasion have been extensively reviewed [4,5,15].

While many different proteins have been shown to play some role in invasion, their precise function during this complex process has almost exclusively been assigned using two assays—end-point invasion assays (Growth inhibition assays GIA), [29] or video microscopy assays [4,30]. End-point invasion assays have been used for decades and have the benefit of throughput and repeatability, but as the readout is whether invasion has occurred or not, it does not provide information about which step in the invasion cascade has been disrupted by a given intervention or gene deletion. By contrast video microscopy, the use of which has expanded in the last decade, can provide information about individual steps during the invasion process; for example it can reveal whether a specific gene deletion/intervention affects the strength of erythrocyte deformation. However, video microscopy involves the passive observation of events and relies mainly on semi-qualitative analysis such as subjective assessment of the extent of erythrocyte deformation or when temporally specific invasion steps start or stop.

Further information on the events of invasion can be gained by using optical tweezers with video microscopy to manipulate merozoites and erythrocytes after egress allowing measurement of the strength of attachment between individual merozoites and erythrocytes [31]. Optical tweezers use focussed beams of light to move objects and are a powerful tool to manipulate biological samples [32–34]. Optical tweezer-based assays have been used previously [35–37] to position merozoites and erythrocytes next to each other and then pull them apart, using well-established measurements of erythrocyte stiffness to convert the extent of erythrocyte stretch during pulling into a measurement of force [37]. This assay has been used to characterise the basic parameters of merozoite-erythrocyte attachment [31] and to assess the impact of erythrocyte polymorphisms on attachment [35,36]. The optical tweezer method, however, has not previously been used to assess the importance of specific *P. falciparum* proteins in merozoite-erythrocyte attachment.

In this study, we combine multiple inhibitors, antibodies, and genetically manipulated *P. falciparum* lines, to dissect the role of individual *P. falciparum* proteins in merozoite-erythrocyte attachment for the first time. In total, we measured 972 detachment forces across all the conditions tested. We found that most proteins could be disrupted or inhibited without significantly impacting the detachment force, including the major merozoite surface protein PfMSP1 (which has been previously proposed to be critical for attachment) and PfGAP45, part of the glideosome actinomyosin motor complex. However, interventions that affect members of the PfEBA and PfRH invasion ligand families, including knockout lines made specifically for this work, did affect the attachment interaction. Overall, the application of the optical tweezers-based attachment assay adds a new function to members of two important *P. falciparum* invasion ligand families.

## Results

### The attachment phenotype of genetically similar merozoites from *P. falciparum* 3D7 and NF54 strain are significantly different

We used optical tweezers to quantitate attachment interactions between newly egressed *P. falciparum* merozoites and erythrocytes, as outlined in more detail in Methods and first established in [31]. In brief, purified schizonts were monitored using video microscopy to identify

schizonts in the process of egress; then optical traps were used to manipulate a newly egressed merozoite and position it between two erythrocytes. After positioning, the cells were left for a period of between 2–5 s to allow for attachment to occur, and then one erythrocyte was pulled with the optical trap, **S1 Video**. As the erythrocyte is pulled, it stretches until the pulling force overcomes the strength of attachment between merozoite and erythrocyte and the cell-cell connection is broken. The stiffness constant of erythrocytes is well-characterised [37], meaning that the cells themselves can be treated like force meters, i.e. the force that was exerted to detach the cells can be estimated from the extent to which the erythrocyte stretches at the point of detachment and hence the strength of the attachment between merozoite and erythrocyte can be calculated **Fig 1A**. The frequency of attachment was also determined by dividing a) the number of merozoites that successfully attached (defined as an attachment strong enough to cause a visual deformation of the erythrocyte when pulled with the optical trap) after positioning between two erythrocytes by the b) the total number of times merozoites were positioned between two erythrocytes from each egressed schizont. This frequency was then averaged across all egresses for a given strain or condition to create a mean attachment frequency **Fig 1B**. All measurements were taken within 180 s of egress to maintain a balance between experimental feasibility and merozoite viability; previous studies with video microscopy have shown that the average time between egress and invasion is 38 s [4] and it has been shown that very few invasions occur after 180 s post-egress [5].

The majority of the experiments in this study were performed using *P. falciparum* strains NF54 or 3D7, or genetically modified lines made using one of these strains as background, so we first set out to establish the attachment interactions of these wildtype lines. NF54 originates from a patient in the Netherlands, and genomic analysis suggests is likely of West African origin [38]. 3D7 is a clone of NF54 that was used for the *P. falciparum* genome project [39] but the two strains have been cultured separately for over 30 years, meaning they have diverged in some aspects of their biology [40–42] while remaining genetically very similar [43]. The optical tweezer assay established that NF54 merozoites had a mean detachment force of 24 ± 2 pN, **Fig 1C** and an attachment frequency per egress of 19 ± 3%, **Fig 1D** (mean ± standard error of the mean is used throughout). By contrast, 3D7 merozoites had a detachment force of 36 ± 3 pN and an attachment frequency of 31 ± 9%. The detachment force of 3D7 was significantly higher than NF54 (t-test p = 0.0007), whereas the attachment frequency was not significantly different (t-test p = 0.13). Given the differences in attachment strength between these two closely related strains, we were interested in whether their growth rates were also different. We set up technical triplicates with tightly synchronised parasites from both lines and measured their growth rates over 6-invasion cycles, see Methods for details. The growth rate was calculated by dividing the parasitemia of the newly invaded rings over the parasitemia measured the day before, giving the parasite erythrocyte multiplication rate (PEMR) as defined by [44]. The PEMR of 3D7 was 2.1 times higher than NF54 (p = ≤0.0000), with NF54 mean PEMR = 4.1 ± 0.3 and 3D7 mean PEMR = 8.7 ± 0.8, **Fig 1E**, reinforcing the fact that there are biological differences between these strains, despite their genetic similarity.

As **Fig 1C** emphasises, there is considerable variability in detachment force between individual merozoite-erythrocyte interactions, even within the same strain. We explored several sources of potential variation within the assay, **S1–S3 Figs**. Firstly, we showed that time since schizont egress had no impact on detachment force, **S1 Fig**. In addition, when sequential detachment forces were measured for the same merozoite, forces could be either higher or lower than previous measurements, **S1F Fig**. We also looked at examples of when two forces were measured for a single detachment using two different erythrocytes; there was only a weak positive correlation between the measurements, indicating a high level of variability. This could be due to measurement error, technical variation such as differences in pulling speed or

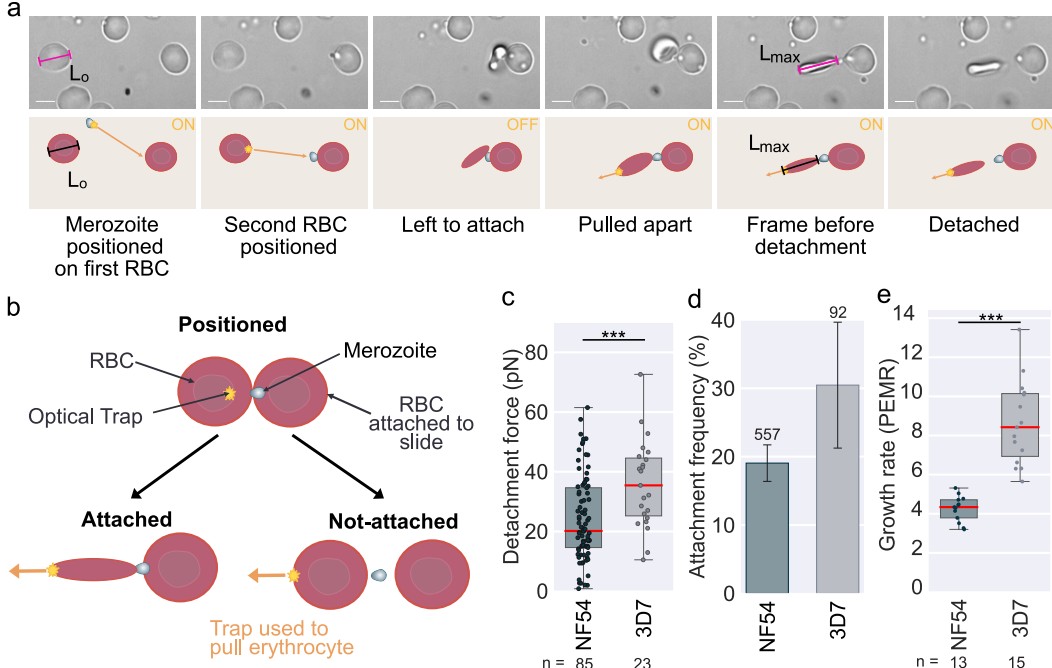

**Fig 1. Optical tweezers show that genetically similar wild-type *P. falciparum* NF54 and 3D7 merozoites have different attachment strengths. (a)** Video-microscopy images from the optical tweezer assay used throughout this work, with a cartoon representation underneath. The orange arrows show the direction of the forces applied to the cells with the optical traps represented as a star. ON/OFF indicates if the optical traps are active during the step. During manipulation, one of the erythrocytes was often attached to the slide, in which case only one erythrocyte was pulled, as in these images. If neither erythrocyte was attached to the slide, then one erythrocyte was held still with an optical trap, and a second trap was used to pull the other erythrocyte. To calculate the force of attachment, the length of the erythrocyte is measured before it is moved ($L_0$), and the maximal stretch of the erythrocyte is measured in the frame immediately before detachment occurs ($L_{max}$). The change in length is calculated as $\Delta L = L_{max}-L_0$. The force of detachment is calculated based on $F = k\Delta L$ [31]. The erythrocyte stiffness constant (k) is assumed to be 20 pN/µm [37]. Scale bar = 5 µm. **(b)** Cartoon to show how the attachment frequency is measured, as described in the text. **(c-e)** Data sets that were significantly different (t-test) have a line between them with stars showing the level of significance *** = ≤0.001. **(c)** Box plot showing the detachment force of merozoite-erythrocyte attachments. NF54 –mean detachment force 24 ± 2 pN (n = 85). 3D7 –mean detachment force = 36 ± 3 pN (n = 23). The difference between strains is significant (p = 0.0007, t-test); both data tests are normally distributed (Anderson-Darling test). The central bold line shows the median, with the top and bottom of the box at the 25th and 75th percentiles and the whiskers showing the total range of the data. At least three biological repeats were completed for both data sets with blood from two different donors. The number (n) of individual interactions measured for each strain is shown under the plot. The attachment behaviour of 3D7 has been previously measured in our lab [35,36]. The detachment force measured previously for 3D7 (43 ± 3 pN), [35], was not significantly different to that measured here (t-test p = 0.16). **(d)** The bar chart shows the frequency of erythrocyte-merozoite-erythrocyte positions that lead to attachment of the merozoite to both erythrocytes. NF54 mean 18.6 ± 2.6% (552 positions) and 3D7 31 ± 9% (92 positions). The number of individual interactions measured for each line is shown above each bar. Error bars show the standard error of the mean (SEM). **(e)** Box plot showing the growth rate measured in static culture represented as the Parasite Erythrocyte Multiplication rate (PEMR), calculated by dividing the parasitemia of the ring stage culture by the parasitemia of the same culture 24 h previously. PEMR was averaged over 6 invasion cycles for three technical repeats per cycle, more details provided in Methods. NF54 mean PEMR = 4.1 ± 0.3 and 3D7 mean PEMR = 8.7 ± 0.8. 3D7 and NF54 were significantly different (p = ≤ 0.0000, t-test). The central bold line shows the median, with the top and bottom of the box at the 25th and 75th percentiles and the whiskers showing the total range of the data.

biological variation, most notably differences in stiffness between individual erythrocytes, **S2 Fig**. Merozoites are polar cells and invasion ligands are secreted at the apical (wide) end, meaning there is likely an asymmetrical distribution of ligands across the surface. To explore whether this meant that there was a preference in which end detachment occurred, we generated a genetically modified line with a fluorescent apical marker, PfAMA1. However, it was not possible to visualise merozoite orientation and detachment at the same time due to the

limited resolution of the microscope and the small size of the merozoites, **S3A–S3D Fig**. Reasoning that attachment might happen preferentially at the apical end; we analysed the wild-type NF54 data to test whether detachment occurred more often from the first or second erythrocyte the merozoite attached to. Detachment did occur slightly more often (68%) from the second erythrocyte the merozoite attached to, but there was no difference in force calculated from either the first or second erythrocyte, **S3E Fig**. These investigations concluded that the assay is robust and quantifiable enough to detect large differences in detachment force (such as between 3D7 and NF54, as in **Fig 1C**) but that multiple potential sources of variability limit the ability to detect more subtle differences.

## Dissecting key *Plasmodium falciparum* determinants of attachment strength

Having established the key parameters of the optical tweezer assay, we next used it with a panel of inhibitors and genetically modified *P. falciparum* parasite lines, each of which disrupts specific steps of the invasion process, as summarised in **Fig 2A**.

### PfMSP1

PfMSP1 is one of the most abundant proteins on the merozoite surface [45]. Evidence for its precise role is conflicting, with several largely indirect pieces of evidence suggesting it is involved in the initial attachment of merozoites to erythrocytes [10,11], perhaps through interaction with band 3 [28] or GYPA [46], while more recent genetic evidence provides strong evidence for PfMSP1 primarily playing a role in egress [27,47]. PfMSP1 is essential for blood-stage growth, so we used a conditional DiCre recombinase-mediated gene excision approach to test its role during merozoite attachment. This conditional system involves integrating two loxP sequences into the target gene in a manner which, when integrated, does not change gene expression but can be recombined to disrupt gene expression upon the addition of rapamycin, which activates DiCre recombinase [48]. A conditional PfMSP1 line has been made previously; however, in this line, PfMSP1 is only truncated [47]. To ensure no fragments of PfMSP1 were left after disruption, which could potentially contribute to binding, we made a new PfMSP1 conditional knock-out line, which after activation, left a modified locus with only a ~16 kDa fragment of PfMSP1, **S4A and S4C Fig**. An excision efficiency of 94% was measured using IFA in rapamycin-treated samples, **S4D Fig**. The efficiency of gene deletion during imaging was assessed by genotyping PCR, which also showed a high efficiency of rapamycin-induced disruption, although some unedited loci were detected in the rapamycin-treated samples. Giemsa-stained smears post-rapamycin treatment showed a mean 2.3-fold reduction in parasitemia compared to DMSO-treated controls, **S5A Fig**, in keeping with previously published data [47]. There was also, as expected, a clear defect in egress characterised by poor release of merozoites seen as clumping of the merozoites (84% in rapamycin compared to 7% in DMSO control), **S3 Video** and **S5B Fig**, again in keeping with previous data [47]. Overall, this suggests a high efficiency of the activation of the conditional knock-out. Optical tweezer measurements of cΔPfMSP1 showed no significant difference in attachment strength between control DMSO treated (30 ± 2 pN) and rapamycin-activated lines (25 ± 2 pN; t-test p = 0.15), **Fig 2B**. The attachment frequency was also not significantly different (rapamycin-treated 22 ± 3%, DMSO-treated control 18 ± 4%; t-test p = 0.42, **Fig 2B**). We also plotted the data using only merozoites that were released from normal egresses for the DMSO-treated sample and the abnormal egresses for the rapamycin-activated knock-out, where we can be confident that rapamycin-induction has deleted the *PfMSP1* gene; there was still no significant difference

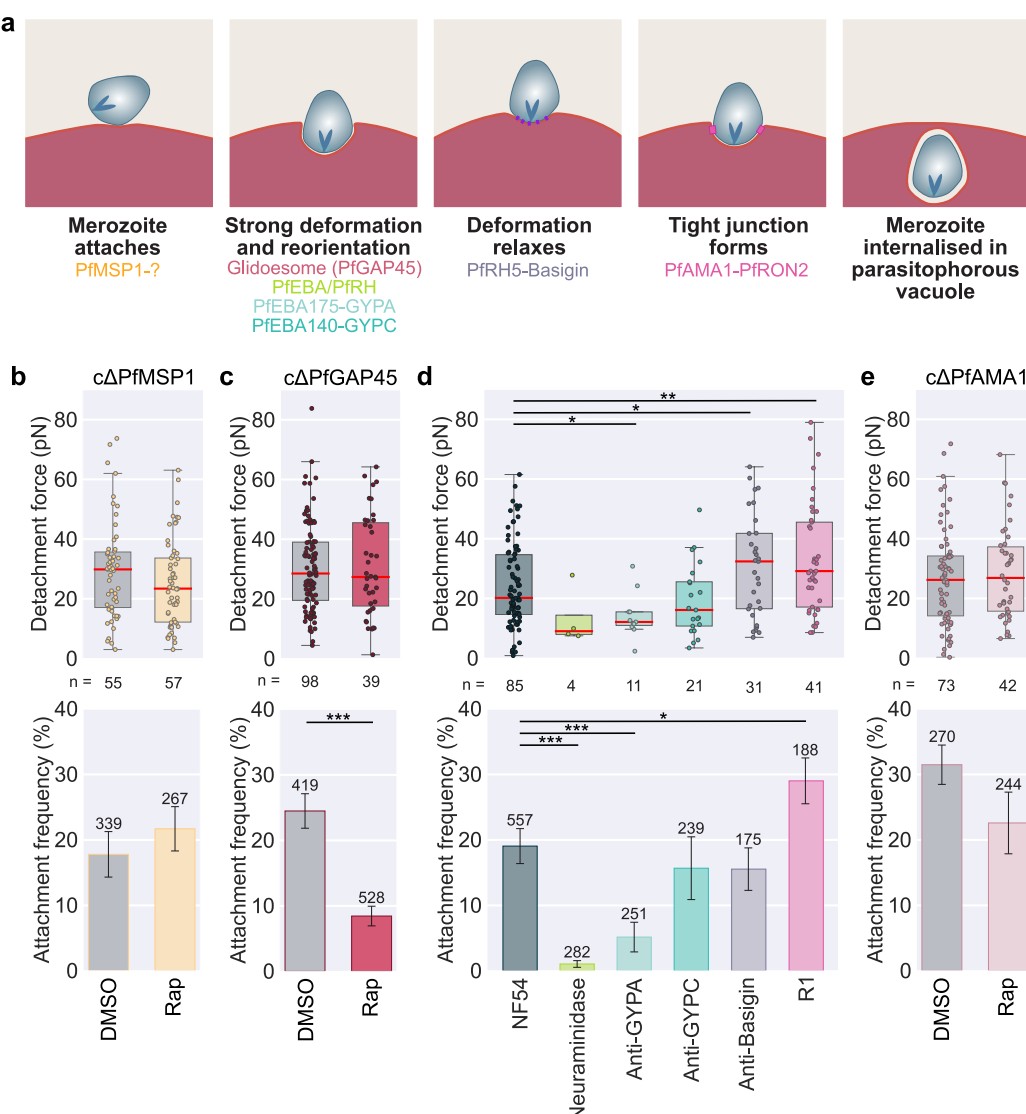

**Fig 2. Impact of disrupting different stages of invasion on merozoite attachment strength. (a)** Schematic summarising the key stages of a merozoite invading an erythrocyte, as well as the stage that each interaction and the inhibitors used to disrupt them are thought to act on. **(b-e)** Data sets that were significantly different (t-test) have a line between them with stars showing the level of significance * = ≤0.05, ** = ≤0.01, and *** = ≤0.001. Box plots showing the detachment force of merozoite-erythrocyte attachment. The central bold line shows the median, with the top and bottom of the box at the 25th and 75th percentiles and the whiskers showing the total range of the data. At least three biological repeats were completed for each condition with blood from at least two different donors. The number (n) of individual interactions measured for each line is shown under the plot. The bar chart shows the frequency of erythrocyte-merozoite-erythrocyte positioning attempts that lead to attachment of the merozoite to both erythrocytes. Error bars show the SEM. **(b, c, e)** were carried out using conditional knock-out lines. DMSO–inactivated control or Rap–rapamycin activated conditional knockout. The number of individual interactions measured for each line is shown above each bar. **(b)** Measurements were carried out using cΔMSP1. **(c)** Measurements were carried out using cΔGAP45. **(d)** Comparison of wild-type NF54 untreated or in the presence of 10 μg/ml of anti-GYPC (BRIC 4), anti-GYPA (BRIC 256) or anti-Basigin (MEM-M6/6); or with neuraminidase treatment 66.7 mU/ml erythrocytes or 20 μM R1 peptide that inhibits PfAMA1-PfRON2 binding. **(e)** Measurements were carried out using cΔPfAMA1.

in detachment force or attachment frequency between the conditions **S5B Fig**. This data, therefore, suggests for the first time that PfMSP1 plays no role in determining merozoite-erythrocyte attachment strength.

## cΔPfGAP45

The glideosome complex, an actinomyosin-based motor complex [22–24], is involved both in the process of erythrocyte entry and in the deformation of the erythrocyte that occurs immediately after initial merozoite attachment [21,25]. We used a conditional knock-out of PfGAP45 (cΔPfGAP45), which has previously been shown to disrupt the assembly of the glideosome and produce merozoites that egress normally but do not deform the erythrocyte membrane on contact and cannot successfully invade [25]. Genotyping PCR after rapamycin treatment confirmed efficient gene excision, in the rapamycin-treated samples, only the (0.7 Kb) bands were seen for the excised locus and no band (1.4Kb) was visible to indicate the presence of unedited parasites and mean parasitemia after reinvasion should have occurred was reduced to 1.4% of the level of the DMSO treated control, **S5C Fig**, in keeping with previous data [25]. Despite this impact on invasion, loss of PfGAP45 caused no effect on the detachment force (DMSO mean– 31 ± 2 pN compared to rapamycin 31 ± 3 pN, not significant p = 0.83) but did significantly reduce attachment frequency (25 ± 2% for DMSO-treated to 8 ± 2% for rapamycin-treated, t-test p = ≤0.0000), **Fig 2C**. This indicates that the glideosome plays a role in parasites attaching/remaining attached for the time of the measurements but does not affect the strength of the subsequent attachment.

## Disrupting PfEBA and PfRH ligand-receptor interactions

After initial attachment has occurred, the merozoite reorientates so that its apical end faces the erythrocyte membrane, and it begins to deform the erythrocyte. This process depends on multiple interchangeable members of the PfEBA and PfRH family of ligands, which bind to an array of erythrocyte receptors, some known and some unknown [15]. We initially used enzymes and antibodies to disrupt known erythrocyte receptor partners to explore the role of PfEBA and PfRH ligands in attachment strength. We confirmed that the antibodies bound to erythrocytes using flow cytometry, **S6A Fig** and replicated their previously reported effects on parasite growth using growth inhibition assays with the NF54 strain **S6B–S6D Fig**.

Neuraminidase cleaves sialic acid residues of glycoproteins and glycolipids on the surface of the erythrocyte. The interactions of GYPA-PfEBA175, GYPC-PfEBA140, GYPB-PfEBL1 (not relevant in NF54 due to a missense mutation in PfEBL1 [49]) and the binding of PfEBA181 and PfRH1 to unknown receptors have all been shown to be sensitive to the neuraminidase treatment [50–52]. Neuraminidase treatment reduced invasion by nearly half in NF54 strain parasites (60 ± 4%), **S6D Fig**, as expected. There was also significantly reduced attachment frequency for neuraminidase-treated erythrocytes (1.4 ± 0.5%), which was significantly different to NF54 (rank-sum p = 0.0001, **Fig 2D**); in fact, only 4 attachments could be measured across 28 egresses and 282 erythrocyte-merozoite-erythrocyte positionings when using neuraminidase treated erythrocytes. For these 4 attachments, the mean detachment force was much lower than for the detachment from untreated blood, but the difference was not significant (13 ± 5 pN, t-test p = 0.14, **Fig 2D**), likely due to the small number of data points that could be achieved due the very low attachment frequency. This strongly suggests that sialic acid-dependent interactions play a major role in merozoite-erythrocyte attachment.

PfEBA175 binds to GYPA on the erythrocyte surface [53,54]. Anti-GYPA antibodies have previously been shown to reduce the invasion efficiency of *P. falciparum* [55], and in our assays, at 10 μg/ml of anti-GYPA invasion was 56 ± 7% that of untreated NF54, **S6B Fig**. The presence of anti-GYPA also caused a significant reduction in the mean detachment force to 14 ± 2 pN, (t-test p = 0.025), **Fig 2D**. As with neuraminidase, the mean attachment frequency per egress 5 ± 2%, was also significantly lower than wildtype NF54 (ranksum p = 0.0002), **Fig 2D**. Inhibition using anti-GYPA therefore has a significant effect on the attachment

behaviour, pointing towards PfEBA175-GYPA binding being a key attachment interaction in the *P. falciparum* NF54 line.

Erythrocyte protein GYPC is the receptor for PfEBA140 [56], anti-PfEBA140 antibody led to a ~20% reduction in invasion in 3D7 [52,56]. Anti-GYPC (BRIC 4) at 10 μg/ml reduced invasion to 89% ± 7% of levels observed in the absence of the antibody, **S6B Fig**, but the same concentration had no significant effect on either detachment force (19 ± 3 pN; p = 0.15, **Fig 2D**), or attachment frequency (16 ± 5%, t-test p = 0.47, **Fig 2D**). This is in keeping with previous evidence that suggested that PfEBA175 is a more critical invasion ligand than PfEBA140 in 3D7 [52], and presumably by inference in NF54, the parent of 3D7 [52]. We also tested anti-CR1 antibodies, as CR1 is the receptor for PfRH4 [57], but saw no significant effect on invasion efficiency, detachment strength or attachment frequency at the concentrations tested, **S6B**, **S6E and S6F Fig**, potentially because the specific antibodies used do not productively interfere with the PfRH4-CR1 interaction.

### Anti-Basigin

As invasion progresses after strong deformation has occurred, PfRH5 binds to Basigin (also known as CD147) [19,30,58]. PfRH5 is the only essential member of the PfRH family [19,59,60] and forms part of the PfPTRAMP–PfCSS–PfRipr–PfCyRPA–PfRh5 (PCRCR) complex [19] which has been proposed to be required for the discharge of the rhoptries and calcium signalling during invasion [17,30]. In keeping with previous studies [58,61], anti-Basigin antibodies at 10 μg/ml reduced invasion to 23 ± 4% of untreated NF54, **S6B Fig**. Anti-Basigin also caused a significant (p = 0.037 t-test) <u>increase</u> in the measured detachment strength, with a mean detachment force of 31 ± 3 pN compared to 24 ± 2 pN for untreated NF54 **Fig 2D**. There was no effect on attachment frequency (16 ± 3% t-test p = 0.44), **Fig 2D**.

### PfAMA1-PfRON2 interaction

The parasite irreversibly commits to invasion when PfRON2 is integrated into the erythrocyte membrane and bound by the PfAMA1 in the merozoite membrane, forming the tight junction [20]. We tested two approaches to disrupt PfAMA1-PfRON2 binding; using a small peptide inhibitor R1 that binds PfAMA1 [62] and a conditional PfAMA1 knock-out line (cΔPfAMA1) [27]. The R1 peptide at 20 μM reduced invasion to 27 ± 3% of untreated parasites **S6C Fig**, in keeping with previously reported figures [20]. As with anti-Basigin antibodies, 20 μM R1 peptide significantly <u>increased</u> detachment force (mean 33 ± 3 pN, t-test p = 0.0037) and it also increased attachment frequency (29 ± 4%, t-test p = 0.055), **Fig 2D**. However, when testing the conditional PfAMA1 knock-out line there was no difference in detachment force (27 ± 2 pN DMSO control vs 29 ± 2 pN rapamycin activated, t-test p = 0.55) or significant reduction in attachment frequency (32 ± 3% DMSO control and rapamycin 23 ± 5%, t-test p = 0.14). Both genotyping PCR and invasion efficiency, **S5D Fig**, showed a high level of editing in the rapamycin-treated samples. There were faint bands in the rapamycin-treated samples for two repeats, indicating that editing was not 100% efficient and there could still be a low percentage of unedited parasites. However, there was a considerable reduction in invasion after rapamycin treatment (mean parasitaemia of rapamycin-treated samples was 2.0% of the parasitaemia of DMSO control samples **S5D Fig**), confirming a high efficiency of successful deletion of PfAMA1 post-rapamycin induction.

### Correlation between invasion rates and optical tweezer measurements

For conditions where we measured the parasite replication rate either through the growth rate assay (NF54 and 3D7) or through the growth inhibition assays (anti-GYPC, anti-GYPA, anti-

Basigin, neuraminidase treatment erythrocytes or R1 peptide) we were interested in how these measurements correlated with the measurements of attachment, **S6 Fig**. When the invasion efficiency relative to NF54 and detachment force were compared there was only a weak positive correlation (correlation coefficient 0.33). However, when we excluded the conditions that disrupted late in invasion (anti-Basigin and R1 peptide), which are involved after the stage likely important to attachment strength, the positive correlation between invasion efficiency and detachment force was much stronger (correlation coefficient 0.982). The relationship between invasion rate and attachment frequency was also strong, with a correlation coefficient of 0.81 when including data from late-stage inhibitors and 0.95 when excluding them. This indicates that there is likely a positive relationship between both the strength of detachment and attachment frequencies with invasion rate, with stronger attachment leading to more productive invasion.

## Understanding the contribution of individual PfEBA and PfRH ligands using single knock-out lines

Several inhibitors that disrupted the interactions of members of the PfEBA and PfRH invasion ligand families (neuraminidase, anti-GYPA and anti-GYPC) reduced the detachment strength in a manner that correlated to a reduction in the invasion rate, **S5 Fig**. The *P. falciparum* PfEBA family consists of PfEBA140, PfEBA175, PfEBA181, PfEBL1 and PfEBA165, although PfEBA165 is a pseudogene in all strains tested to date [63,64], and PfEBL1 is also a pseudogene in multiple strains, including the NF54 background used here [49]. The *P. falciparum* PfRH family consists of PfRH1, PfRH2a, PfRH2b, PfRH3, and PfRH4 (and PfRH5, although as noted above it is thought to operate at a later invasion step, and was not considered further here). Like PfEBA165, PfRH3 is also a pseudogene in all strains tested to date [65].

In order to test more systematically the role in attachment of specific proteins in the PfEBA and PfRH family we generated a panel of knock-out lines, **Fig 3A**. All PfEBA and PfRH knockouts have been individually made previously: 3D7ΔPfEBA175 [66,67], W2mefΔPfEBA175 [66], 3D7ΔPfEBA181 [67], 3D7ΔPfEBA140 [56,67], W2mefΔPfEBA140 [56], Tak994ΔPfRH1 [68], 3D7ΔPfRH1 [69], 3D7ΔPfRh2a [69], 3D7ΔPfRh2b [52,69] and W2mefΔPfRH4 [70], but these knockouts exist on a wide range of strain backgrounds and were generated using different experimental genetic approaches. To allow for a more complete comparison, we assembled a consistent set of knockouts constructed in the same manner in NF54 background. Two control knock-out lines were made using the same approach targeting PfP230P and Pfs25; both proteins are gametocyte specific, have very low-level expression in the asexual stage and have no known role in invasion [71–73]. Some knockouts (NF54ΔPfEBA140, NF54ΔPfEBA181, NF54ΔPfRH4, NF54ΔPfEBA175, Dd2ΔPfEBA175) were made as part of a larger systematic study of *P. falciparum* genes, while others (NF54ΔPfRH2a, NF54ΔPfRH1, NF54ΔPfP230P, NF54ΔPfs25) were made specifically for this study. All lines were cloned by limiting dilution and genotyped following cloning specifically for this study **S8A and S8B Fig**, see Methods for further details.

## Effect of PfEBA and PfRH gene deletions on growth

We began by carrying out a growth assay on the knock-out lines, **Fig 3B**. We used tightly synchronised parasites in triplicates and sampled parasitemia daily using flow cytometry over multiple invasion cycles (see Methods for further details). The Parasite Erythrocyte multiplication rate (PEMR), [44], was then calculated by dividing parasitemia after invasion by parasitemia in the same culture the day before. Under static culture, there were significant differences in the mean growth rate relative to NF54 (mean PEMR = $4.5 \pm 0.3$) for NF54ΔPfs25

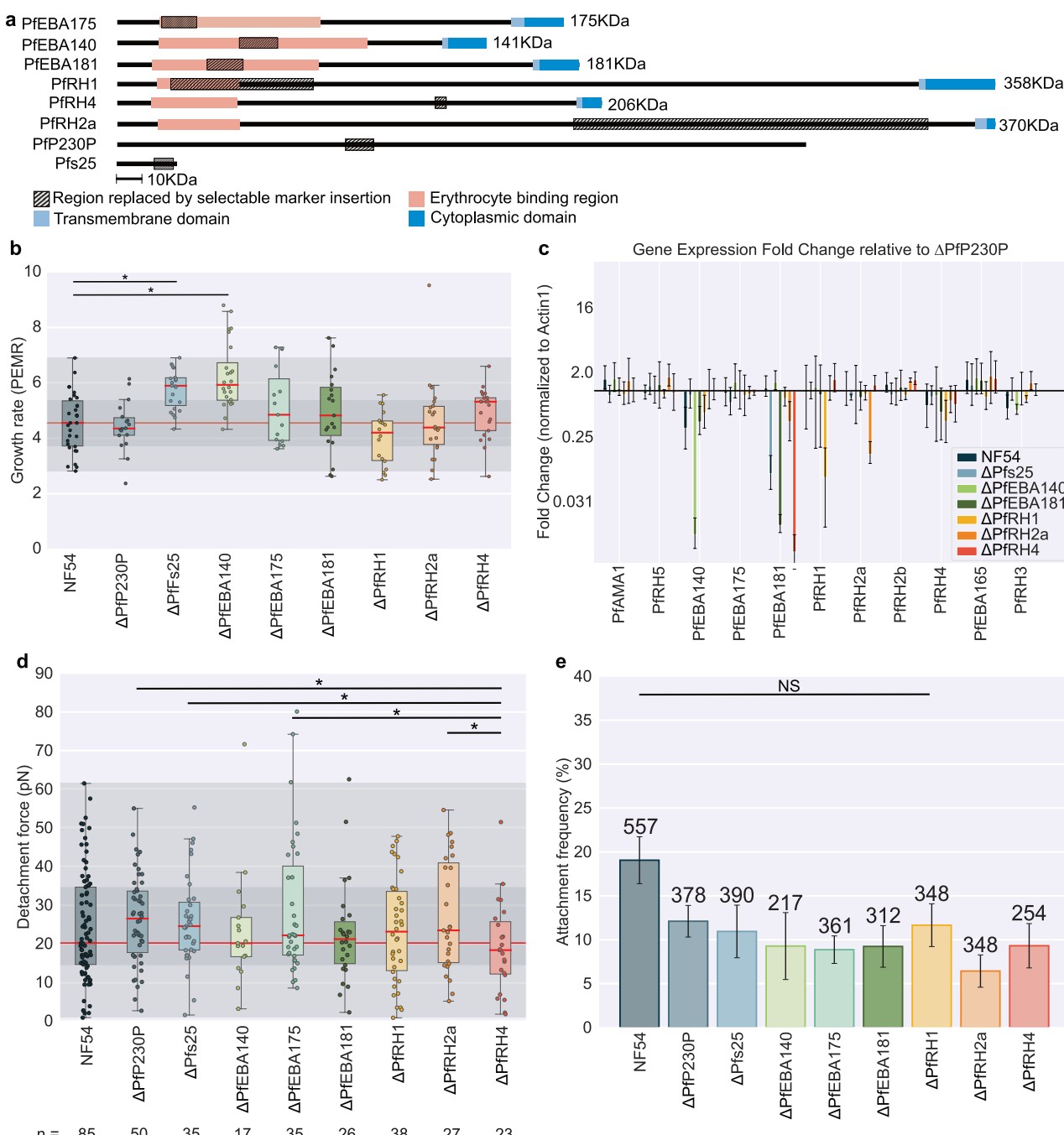

**Fig 3. Characterisation of PfEBA and PfRH knock-out lines (a)** Schematic of the regions disrupted in each knock-out line. Striped regions indicate the regions deleted and replaced by the drug resistance cassette. For all genes, this insertion was made within the predicted erythrocyte binding domain apart from in PfRH4, which was generated as part of another study, and PfRH2a, where the binding domain is identical between PfRH2a and PfRH2b, meaning a unique deletion must be targeted elsewhere. The pink regions show the predicted binding regions, which were predicted using https://www.ebi.ac.uk/interpro. The light blue regions show the predicted transmembrane domains and the dark blue region shows the cytoplasmic domains. **(b, d-e)** Data sets that were significantly different (t-test) have a line between them with stars showing the level of significance * = ≤0.05, ** = ≤0.01, and *** = ≤0.001. The central bold line shows the median, with the top and bottom of the box at the 25th and 75th percentiles and the whiskers showing the total range of the data. The red line across the graph shows the median and shading across the whole graph shows the 25th to 75th percentile range of NF54. All knock-out lines tested are in the NF54 background. **(b)** Box plot showing the growth rate measured in static culture represented as the Parasite Erythrocyte Multiplication Rate (PEMR), calculated by dividing the parasitemia of the ring stage culture by the parasitemia of the same culture measured 24 h previously. PEMR was measured as detailed in the Methods. At a 5% level of significance, the only lines that were different to NF54 mean PEMR = 4.5 ± 0.3 were NF54ΔPfs25 PEMR = 5.8 ± 0.2 (p = 0.0025) and NF54ΔPfEBA140 PEMR = 6.3 ± 0.4 (p = 0.0020). **(c)** Analysis of invasion-associated gene expression using qPCR. Error bars show the standard deviation between repeats. The expression in each sample is

normalised to the housekeeping gene PfActin I and stars indicating significant changes at greater than a 5% significance level (t-test). The gene expression of each knock-out line is expressed as a relative fold change compared to NF54ΔPf230P. We monitored the expression of the PfEBA and PfRH family of proteins, as well as two essential invasion genes PfAMA1 and PfRH5. Note that ΔPfEBA175 was not included in this experiment as the line was not available at the time. (**d**) Box plots showing the detachment force of merozoite-erythrocyte binding. The central bold line shows the median, with the top and bottom of the box at the 25th and 75th percentiles and the whiskers showing the total range of the data. The red line across the graph shows the median and shading across the whole graph shows the 25th to 75th percentile range of NF54. At least three biological repeats were completed for each data set with blood from two different donors; the number (n) of individual interactions measured for each line is shown under the plot. (**e**) The bar chart shows the frequency of erythrocyte-merozoite-erythrocyte positions that lead to successful attachment of the merozoite to both erythrocytes. The attachment frequency of all the knockouts was not significantly lower than NF54 at p = ≤0.05 except NF54ΔPfRH1 which was nearly significant at a 5% level (t-test p = 0.058), data sets that were not significantly different (t-test) have a line between them with NS above. The number of erythrocyte-merozoite-erythrocyte positioned for each line is shown above each bar. Error bars show the SEM.

(PEMR = 5.8 ± 0.2) was 28% higher (p = 0.0025) and NF54ΔPfEBA140 (PEMR = 6.4 ± 0.4) was 38% higher (p = 0.0020). The fact that the growth rate of NF54ΔPfs25 was higher than NF54, despite Pfs25 having no known role in invasion, suggests that the invasion rate can change during the process of generating the knockout lines. Differences in invasion phenotypes of clones of the same line have been reported previously [74–77]. The growth rate of NF54ΔPfEBA140 was not significantly different to NF54ΔPfs25 (p = 0.28), so we cannot conclude that this difference is due to PfEBA140 deletion.

### RT-qPCR shows few significant changes in invasion gene expression in PfEBA and PfRH knockout lines

PfEBA and PfRH proteins are known to have variable expression patterns which has been correlated with invasion phenotype changes [67,70,78–82]. Generation of a cloned knock-out line requires 3–6 months of culturing, so we assessed whether any of our PfEBA and PfRH knock-out lines also carried transcriptional changes that could impact phenotype by using qPCR to compare transcript levels across all family members. PfEBA and PfRH proteins have highly stage-dependent expression, with expression peaking in late schizonts (~44–48 h) and merozoites [81,83,84]. Samples were therefore collected of tightly synchronised schizonts treated with Compound 2 (C2), which pauses the schizonts ~15 mins before egress [85]. We assessed whether the C2 treatment had any transcriptional impact by measuring the expression of the invasion genes in wild-type NF54 before and after 3.5 h of C2 treatment; no differences were observed, **S9A Fig**. For most of the PfEBA/PfRH knockout lines, there was a large decrease in transcription of the targeted gene (NF54ΔPfEBA140, NF54ΔPfEBA181, NF54ΔPfRH1, NF54ΔPfRH2a), as expected (**Figs 3C and S9B**). The only exception was NF54ΔPfRH4, where there was no significant change in PfRH4 expression relative to NF54ΔPFP230P. However, the qPCR primers amplified a region downstream of the integrated selectable marker for all knock-outs, except for NF54ΔPfRH4 where they amplified a region upstream of the integration, meaning they may detect a stable yet truncated transcript. Other than the expected decrease in expression of target genes, there were no consistent differences in expression patterns across lines, but expression levels of PfEBA181, PfEBA140 and PfRH1 did frequently differ between lines, even in the control knockouts, and in NF54ΔPfRH2a there was a small but significant 1.4-fold increase in the expression of PfRH2b (t-test p = 0.034). The ΔPfEBA175 line was not tested as it was not constructed when this experiment was carried out.

### Effect of PfEBA and PfRH gene deletions on merozoite-erythrocyte attachment strength

Optical tweezer measurements were performed on all PfEBA and PfRH knock-out lines along with the control lines. Across all knockout strains, 267 egresses were observed, 2608 individual erythrocyte-merozoite-erythrocytes were positioned, and 251 attachments were measured,

Fig 3D and 3E. The attachment frequency of all the knockouts (except NF54ΔPfRH1) was significantly lower than NF54, including both controls NF54ΔPfP230P (t-test p = 0.049) and NF54ΔPfs25 (t-test p = 0.045), Fig 3E. This indicates the process of constructing the knockout lines impacted the resultant attachment frequency, but there was no difference in attachment frequency between knockout lines. By contrast, the detachment force of NF54ΔPfRH4 (mean 19 ± 3 pN) was significantly lower than both NF54ΔPfs25 (mean 26 ± 2 pN p = 0.0321) and ΔPfP230P (mean 26 ± 2 pN p = 0.044) control lines, Fig 3D, suggesting a role for PfRH4 specifically in attachment strength in NF54. None of the other knockout lines had significantly different distributions of detachment force compared to NF54 or the two control lines.

The lack of phenotype in the NF54ΔPfEBA175 line is very different from the significant decrease in detachment force after treatment with anti-GYPA, which targets the same PfEBA175-GYPA interaction albeit through a different inhibitory mechanism. This interaction is thought to be more important for invasion in the Dd2 strain than in NF54 because invasion in Dd2 is almost completely blocked by neuraminidase treatment, whereas in NF54 and 3D7 neuraminidase treatment only reduces invasion by around 50%, S6D Fig and [86–88]. We therefore also tested a ΔPfEBA175 line made in the Dd2 background. The optical tweezers attachment assay showed unexpectedly that Dd2ΔPfEBA175 had a significantly (p = 0.0001) higher detachment force (mean = 31 ± 2 pN) compared to the Dd2 wildtype control (mean = 23 ± 2 pN), S10B Fig, whereas the attachment frequency was significantly lower, in keeping with the knockouts made in the NF54 background, Fig 3E. While surprising, it is notable that PfRH4 has been shown to be significantly up-regulated when PfEBA175 is disrupted in the parental lines of Dd2 W2mef [70]–there is therefore consistency between the reduction in detachment force when PfRH4 is deleted, and an increase in detachment force when it is up-regulated, implying that PfRH4 may play a particularly important role in attachment. We next tested the Dd2ΔPfEBA175 line in the presence of anti-GYPA, which resulted in a significant reduction in detachment force (mean 16 ± 2 pN, p = 0.0001) compared to Dd2ΔPfEBA175. This indicates that some of the effect of anti-GYPA inhibition is independent of disruption of the PfEBA175-GYPA interaction, S10B Fig, either acting on other PfEBA interactions or in another indirect manner.

## Discussion

In this study, we used optical tweezers to systematically test the impact of a range of inhibitors and gene deletions on *Plasmodium falciparum* merozoite-erythrocyte attachment efficiency and strength, as summarised in Fig 4. The forces measured here range from 1 pN to 84 pN depending on the different specific strain and condition. Optical tweezers have previously been used to assess the force with which *Plasmodium berghei* sporozoites bind to glass slides and beads (20–190 pN) [89–91] and the force with which *Toxoplasma gondii* could move a microsphere attached to its surface [92], making the merozoite-erythrocyte attachment forces broadly comparable to the attachment behaviour of these other apicomplexan zoites. For context, a typical single-molecule interaction ranges from ~1.7 pN for actin-myosin to ~160 pN for biotin-streptavidin [93], while the forces measured for erythrocytes binding to infected erythrocytes in rosettes are several orders of magnitude stronger (44 nN) [94]. Emphasising that the measure we were quantifying is significant in the context of the invasion process, mean detachment force showed a strong positive correlation with parasite growth rate in NF54/3D7, of which invasion efficiency is a key component, S6 Fig. The correlation between detachment force and growth is not as strong if data from inhibitors that disrupt late in the process of invasion (anti-Basigin and R1 peptide) are included, S6 Fig. This makes biological sense as if invasion is blocked after attachment has occurred, then invasion rate would be

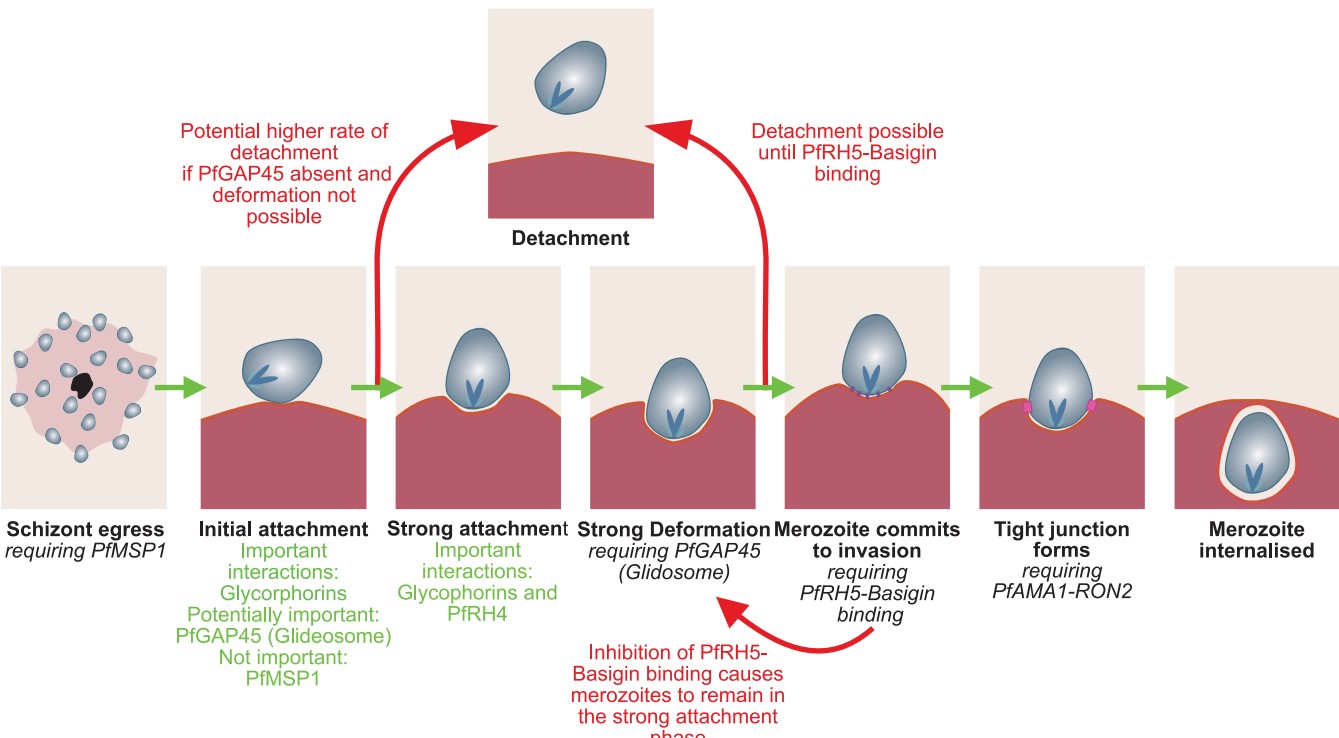

**Fig 4.** Schematic summarising the findings on attachment interactions discussed in this paper (shown in red and green) in the context of the latest understanding of the model of *Plasmodium falciparum* invasion (black). Green arrows show the progression of the merozoite through the sequence of invasion. Red arrows indicate the merozoite reversing the progress of invasion or the detachment of the merozoite.

reduced, but attachment behaviour would remain unchanged. The link between detachment force and growth is also emphasised when the two wild-type strains tested, NF54 and 3D7 are compared– 3D7 had a significantly higher detachment force (3D7–36 ± 3 pN vs NF54 24 ± 2 pN, t-test p = 0.0007), **Fig 1C** and 2.1 times higher growth rate **Fig 1E**. We therefore believe that measuring merozoite-erythrocyte detachment force, which we have done systematically here for the first time, is biologically relevant and provides important new information about the invasion process.

The comparison of wild-type strains also emphasizes that attachment strength is not only controlled by genetic factors. 3D7 is a clone of NF54 [38]; genetic comparison has shown that over the 30 years in which the two lines have been cultured independently, they have gained very few genetic differences, none of which were found in proteins linked to invasion, [43]. There have, however previously been several differences in biological phenotype reported between 3D7 and NF54: in *var* gene expression [41], gametocyte conversion rate [40] and heterochromatin structure [42]. Therefore, epigenetic differences between the lines may cause differences such as in the expression of invasion ligands [95,96] and may underpin the differences in attachment behaviour and growth rates we observed. Even different clones of 3D7 have been reported to have very different invasion phenotypes [74–76], emphasising the importance of non-genetic differences in invasion phenotypes.

The number of detachment events measured in this study represents a 20-fold increase over data gathered previously [31], allowing us to investigate for the first-time technical factors that could influence the measured detachment force. We have shown that the detachment force of the merozoite does not change within the first 180 s of egress and that the same merozoite can

attach multiple times without a drop in detachment force, **S1 Fig**, suggesting either that ligands can bind reversibly and be reused multiple times, or that each merozoite has a large pool of ligands able to attach to merozoites. Whilst we were not able to determine the polarity of the detachment using an apically localised fluorescent marker during the optical tweezer measurements, we were able to show that the merozoites do have a slight preference for detaching from the second erythrocyte they attached to (68% of the time). The merozoite will presumably reorientate its apical end towards the first erythrocyte it contacts, so the more frequent detachment from the second erythrocyte suggests that merozoites may form slightly more robust attachments at the apical end, but this is not a hypothesis that we were able to directly quantitate in this study.

This analysis does reveal that the optical tweezers method has some limitations; in particular, the measurements of detachment forces have high variability, **S2 Fig**. This is likely largely due to the variation in the stiffness of erythrocytes, which is assumed to be constant in the calculation process whereas there is known to be at least 20% variation across cells within even a single donor [37]. Some of the spread in forces measured may also be due to biological variation between ligands on the individual erythrocytes/merozoites, for example, erythrocyte age can affect the abundance of specific surface receptors and there is also evidence that within a population of merozoites, there is variation in invasion ligand expression [97]. Together, this means that the method is likely only able to significantly distinguish large differences in force and needs sufficient data points to do so. Collecting large data sets, however, is challenging due to the highly technical and manual nature of collecting and analysing measurements, with biological repeats of a single intervention requiring multiple weeks of work.

A key goal of this work was that it allowed us to directly assess the impact of specific *P. falciparum* proteins on merozoite attachment for the first time. At the outset of the work, one prediction was that the major merozoite surface protein PfMSP1 could play a key role in attachment strength, as there are several indirect pieces of evidence suggesting it is involved in the initial attachment of the merozoite to erythrocytes [10,11,28,46]. However, at least part of this hypothesis has been inferred from the effects of heparin treatment, which is a highly anionic sulphated polysaccharide and while there is some evidence that it is a selective inhibitor of PfMSP1 function, it is possible that it has a broader inhibitory effect. Using a conditional PfMSP1 deletion line to ensure specificity, our data clearly showed no significant change in attachment frequency or detachment force in the absence of PfMSP1, **Fig 2B**, providing conclusive evidence that the primary role of PfMSP1 is not in attachment during invasion [27,47]. This also indicates that weak interactions formed by PfMSP1 are not required to facilitate later stronger interactions by other ligands as, if this were the case, we would also expect to detect changes in the detachment forces or attachment frequency. We cannot rule out that PfMSP1 has a very weak contribution to attachment strength, below the sensitivity of our assay to measure. Similarly, deletion of PfGAP45, an essential part of the glideosome actinomycin-based contractile system [23], also had no impact on attachment strength, although it did significantly reduce attachment frequency, **Fig 2C**, perhaps linked to the fact that video microscopy has previously shown that merozoites lacking PfGAP45 could not deform erythrocytes as normally seen during the initial phase of invasion [25]. If a merozoite cannot undergo deformation, it may be more likely to detach as it cannot progress to the later steps of invasion.

Instead, several lines of evidence suggest that interactions of several of the PfEBAs/PfRHs are key to merozoites attaching/remaining attached to erythrocytes. Firstly the attachment frequency to neuraminidase-treated erythrocytes (which removes the sialic acid residues on erythrocyte surface proteins) was just 1.4 ± 0.5%, **Fig 2D,** and the binding interactions of PfEBA175, PfEBA140, PfEBA181, PfEBL1 and PfRH1 have all been shown to be neuraminidase sensitive [50–52]. Interestingly, despite the very low attachment frequency, the invasion

rate into neuraminidase treated erythrocytes was still 57 ± 3%. Invasion requires successful attachment to one erythrocyte, but in this assay, successful attachment is defined as an attachment to two erythrocytes, which may increase the apparent effect of neuraminidase treatment on attachment relative to its effect on invasion. In our assay attachment also must be strong enough to be visually detectable by video microscopy, meaning we may miss weaker attachments that are still strong enough to allow invasion to occur in static *in vitro* culture conditions, when merozoites and erythrocytes sit next to each other for an extended period of time. Secondly, disruption of PfEBA175-GYPA binding via anti-GYPA, **Fig 2C**, significantly reduced the attachment frequency to 5 ± 2%, significantly reduced both detachment force and attachment frequency, although the discordance between the effect of anti-GYPA and deleting PfEBA175 in NF54 and Dd2 suggests that at least some of the impact of the anti-GYPA antibodies may be indirect. Finally, deleting PfRH4 reduced attachment strength compared to control lines NF54ΔPfP230P and NF54ΔPfs25, while deleting PfEBA175 in Dd2, which is known to increase PfRH4 expression, increased attachment strength. While a direct role for PfRH4 in attachment strength has not been proposed before, PfRH4 gene expression has been reported to be upregulated when Dd2 parasites are cultured under shaking conditions, where stronger attachment might be hypothesised to be more beneficial to increase invasion efficiency in the face of increasing shear forces [98,99]. PfRH4 binds CR1, [57], and while we were not able to detect an effect on attachment or invasion when using commercial anti-CR1 antibodies, we had no assay available to confirm that these antibodies were functionally disrupting PfRH4-CR1 binding. Across all conditions tested, neuraminidase-treated erythrocytes had the lowest mean detachment force, although the difference was insignificant as only 4 measurements were taken. This suggests that glycophorin binding interactions, along with PfRH4 binding, are likely most important for binding strength in NF54.

The disruption of several other PfEBA/PfRH proteins did not affect the detachment force with no significant differences measured with anti-GYPC, NF54ΔPfEBA181, NF54ΔPfEBA175, NF54ΔPfEBA140, NF54ΔPfRH1 and NF54ΔPfRH2a. Either these ligands form weaker interactions than, for example, PfRH4 (which could be dominant if expressed), or they are not preferentially utilised in the NF54 strain tested, so they have a minimal role in attachment. Given the known variability in gene expression in these families and the length of time it takes to create cloned knockout lines, it is also possible that the knock-out lines had changed expression of invasion ligands during the extended culture period. However, quantification of key invasion genes expression with RT-qPCR, **Figs 3C and S9B** showed very little change in expression in the knock-out lines other than in the targeted gene, except for downregulation of PfEBA181 in NF54ΔPfs25 and NF54ΔPfRH4. The lack of transcriptional adaption suggests that some ligands are dominant in determining attachment strength either because they have intrinsically stronger interactions or because they are preferentially utilised in a given strain. An interesting avenue for future work would be to characterise knock-outs in other wild-type backgrounds and to test double PfEBA/PfRH knockout lines to see whether there is a cumulative effect of deleting multiple PfRH and PfEBA ligands on attachment.

Inhibiting later steps in invasion using anti-Basigin and the R1 peptide resulted in an increase, rather than a decrease, in attachment strength. While this might initially seem counter-intuitive, it is consistent with video microscopy studies that show inhibition of PfRH5-Basign binding [17,19,30] or PfAMA1-PfRON2 interaction [30,100] show a longer average duration of erythrocyte deformation. This suggests that merozoite-erythrocyte interactions persist for longer in these conditions, and merozoites may be stuck in a strongly attached/strong deformation phase. The attachment frequency and detachment force conflicted in the two ways we disrupted the PfAMA1-PfRON2 interaction. There was no significant change in attachment frequency or detachment force when the PfAMA1 gene was

conditionally deleted, whereas there were significant increases in both when the R1 peptide, which is thought to inhibit the same interaction, was present. The R1 peptide has been shown to bind to the hydrophobic grove of PfAMA1 that binds to PfRON2 blocking binding [101], but there is evidence that there is an additional binding interaction of PfAMA1 with a second region of PfRON2 or an unknown erythrocytic receptor [102]. This could explain the difference in phenotype between the R1 peptide, which will only affect one interaction, and the cΔPfAMA1 strain, where all PfAMA1 interactions will be inhibited. [27]

To summarize, we have combined optical tweezers with a range of inhibitors and genetically modified lines to systematically study the function of *P. falciparum* invasion proteins during attachment for the first time, **Fig 4**. We have provided evidence that PfMSP1 does not have a significant role in merozoite attachment. This is in keeping with PfMSP1 having an essential role in egress [27,47], but our data alone cannot completely rule out a minor role in either attachment or invasion; further studies using the conditional PfMSP1 deletion line will be needed to explore PfMSP1 function more fully. Attachment frequency depends on a functional glideosome, but attachment strength does not, suggesting attachment strength is not dependent on erythrocyte deformation, which has previously been shown to be driven by the glideosome [21,25]. Evidence from both inhibitors and genetically modified lines suggests that the strength of merozoite-erythrocyte attachment is primarily dependent on the PfEBA and PfRH invasion ligand superfamilies. Redundancy in function between members of these families makes dissecting the contribution of individual proteins challenging, but in the NF54 background the interactions of PfRH4 appear more important to the strength of attachment than PfEBA140, PfEBA175, PfEBA181, PfRH1 and PfRH2a. By contrast, interactions involved in the final stages of invasion, such as binding of the PCRCR complex, central to which is the PfRH5-Basign interaction [17,58], occur at a point at which the merozoite is committed to invasion and can no longer readily detach [18,19], and consequently are not primarily involved in attachment strength. Finally, we reveal an intriguing correlation between merozoite-erythrocyte attachment strength and parasite growth rate. While invasion is one of the most closely studied biological processes in the *P. falciparum* erythrocytic cycle, most studies of invasion have been carried out *in vitro* in static conditions which are likely very different to the turbulent *in vivo* environment where both merozoites and erythrocytes are subject to flow forces by the circulatory system. Tools such as the optical tweezers used here, as well as new approaches such as microfluidics, have the potential to shed new light on this essential and much-studied process.

## Materials and methods

### Ethics statement

Ethical approval for the use of human blood was obtained from National Health Service (NHS) Cambridge South Research Ethics Committee, 20/EE/0100, and the University of Cambridge Human Biology Research Ethics Committee, HBREC.2019.40, formal written consent was obtained at the point of collection and is held by the NHS Blood and Transplant (NHSBT).

### *Plasmodium falciparum* culture

*P. falciparum* strains (either the background NF54, 3D7 or Dd2 lines, or genetically modified lines described below) were cultured in human erythrocytes (purchased from NHSBT, Cambridge, UK) at a 4% haematocrit (HCT) in RPMI 1640 medium (Gibco, UK) supplemented with 5 gl$^{-1}$ Albumax II, DEXTROSE ANHYDROUS EP 2 g l$^{-1}$, HEPES 5.96 g l$^{-1}$, sodium bicarbonate EP 0.3 gl$^{-1}$ and 0.05 gl$^{-1}$ hypoxanthine dissolved in 2 M NaOH. Culture was performed

at 37˚C in a gassed incubator or gassed sealed culture container under a low oxygen atmosphere of 1% $O_2$, 3% $CO_2$, and 96% $N_2$ (BOC, Guildford, UK).

## Sorbitol synchronisation

Sorbitol synchronisation was done as described by [103]. Ring stage cultures were pelleted by centrifugation, the supernatant was removed, and the pellet was resuspended in ten times the pelleted volume of 5% D-sorbitol (Sigma-Aldrich), then incubated at 37˚C for 5 mins, during which time later stage parasites (trophozoites and schizonts) rupture. The cells were centrifuged and resuspended in RPMI at twenty times the pelleted volume. The cells were pelleted again and then resuspended in complete medium to give 4% HCT.

## Percoll synchronisation

Smears were checked to confirm predominantly schizonts and a few rings. Most of the media was removed and the infected blood was resuspended in 5 ml of media, then layered over 5 ml of 70% Percoll in a 15 ml tube (Percoll Merck P4937, 10% 10x PBS, 20% RPMI). The tube was then centrifuged at 1450 rcf for 11 mins with the break set to 1 and accelerator to 3. The band of late-stage parasites at the percoll-media interface was then added to a flask with media and blood at a 4% HCT and incubated in a gassed incubator at 37˚C for 3.5 h unless indicated otherwise. The Percoll separation was then repeated as above but this time the bottom pellet containing ring stage parasites (which had invaded in the previous 3.5 h) was kept and then sorbitol synchronised as described above.

## Optical tweezers imaging

The method used to carry out the optical tweezers analysis is described in [31]. Protocol available DOI: dx.doi.org/10.17504/protocols.io.5qpvo3zmzv4o/v1. Cultures were sorbitol synchronised at least three times in consecutive cycles before imaging; a 25 ml culture was split so 20% of the flask was diluted for continuous culture and the remainder was used for imaging. Schizonts were isolated for imaging using magnetic purification [104] with a magnetic cell fraction column (MACS, Miltenyi Biotec) and a MidiMACS separator. The eluted schizonts were pelleted by centrifugation and resuspended in 200–800µl of complete medium, depending on the pellet size. Uninfected blood was added to 200ul of schizonts to give a 0.025% HCT. If antibodies, **S1 Table**, or the R1 peptide were added, they were added at the indicated concentrations at this stage and were present for the imaging duration. Imaging was performed with the cells in a SecureSeal hybridisation chamber (SecureSeal; Grace Bio-Labs, Bend) and coverslips coated with 6 µl of poly(l-lysine)-graft-poly(ethylene glycol) (PLL [20]-g[3.5]-PEG [2]) (SuSoS AG, Dubendorf, Switzerland) at 0.5 mg/mL concentration to reduce adherence of the erythrocytes to the coverslip.

The temperature was kept at 37˚C throughout the experiment using a custom-built temperature-control stage. Imaging was performed on an Eclipse Ti-E, Nikon inverted microscope and a 60x water objective (Plan Apo VC 60x 1.20 NA water objective, Nikon). The custom-built optical tweezers set-up, described in [105], consists of a solid-state pumped Nd:YAG laser (PYL-1-1064-LP, 1064 nm, 1.1 W (IPG Laser). The traps are steered with a pair of acousto-optical deflectors (AA Opto-Electronic, Orsay, France) controlled by custom-built electronics; multiple traps can be used with subnanometer position resolution. Videos are recorded in brightfield at 20 frames/s using model Grasshopper3 GS3-U3-23S6M-C (Teledyne FLIR) with a Physical pixel size of 5.86 µm resulting in a 0.0973 µm pixel size in the images. The cells were manipulated as indicated by **Fig 1A**. Three repeats were done for each condition

on separate days using at least two different donors' blood, to account for both human and parasite potential sources of biological variation.

## Determining the attachment frequency

The frequency of attachment was established for each condition by calculating the frequency at which an adhered erythrocyte-merozoite-erythrocyte chain was formed when two erythrocytes were placed next to a merozoite and left for ~2–5 s **Fig 1B**. If the same combination of cells was positioned more than three times, they were no longer counted. Measurements were only recorded for erythrocyte-merozoite-erythrocyte chains positioned within 180 s of egress.

## Determining the detachment force

Pulling is done in one of two ways, depending on whether one erythrocyte is attached to the glass slide. If one erythrocyte has attached to the slide, then only one trap is needed and is used to stretch the other erythrocyte (for NF54 this occurred 78% of the time). If neither erythrocyte is attached to the slide, two traps are used, one to hold one erythrocyte, while the other erythrocyte is pulled using a second trap. It has been shown that erythrocytes' mechanical properties can be modelled as a linear spring [37]. This means there is a linear relationship between the force applied and the stretch of the erythrocyte along a single axis. A healthy human erythrocyte has a stiffness (k) of 20 pN/μm [37] with ~20% uncertainty due to erythrocyte variation. The elongation of the erythrocyte ($\Delta L = L_{max}-L_0$) was calculated by measuring the length of each erythrocyte before it was moved ($L_o$) and subtracting it from the length in the frame before detachment I ($L_{max}$), **Fig 1A**. Measurements of length were performed using ImageJ software. The force of detachment is calculated based on $F = k\Delta L$ [31]. Measurements were only taken if the attachment was by a single merozoite, and force measurements were not taken from erythrocytes attached to the glass slide. If one of the erythrocytes undergoes echinocytosis then it is not measured. If two erythrocytes were stretched, then one force was selected at random and included in the data set (so that two measurements for the same detachment were not included).

## Growth rate assay

At the start of the assay all lines tested were defrosted at the same time; they were sorbitol synchronised in the first cycle when >0.5% rings and then delayed at room temperature to push the egress window to the intended start window. The percoll synchronisation described above was carried out with a 3 h invasion window. The parasitemia was then counted using Giemsa smears and the cultures diluted to a 0.2–0.5% parasitemia at a 4% HCT (the parasitemia was set as high as possible up to 0.5% based on the number of parasites available and the number of wells). For the duration of the assay, the lines were grown in 5 ml at 4% HCT in a 6-well plate. Every day at ~3 h after the initial invasion window the parasitaemia was measured using flow cytometry. 5 μl of each well was diluted in 45 μl of PBS in a 96-well plate. The samples were stained using SYBR Green I (Invitrogen, Paisley, UK) at a 1:5000 dilution for 45 mins at 37°C. The samples were then run on an Attune NxT acoustic focusing cytometer (Invitrogen) with SYBR green excited with a 488 nm laser (BL1-A) and detected by a 530/30 filter. The parasitemia was measured using the Attune NXT software; for each data set, a plot of SSC-A vs FSCA was used to gate for roughly the size of an erythrocyte, and then singlets were gated for in an FSC-H vs FSC-A. Next histograms for the BL1-A intensity were used to gate for the SYBR green positive erythrocytes; the percentage of positive erythrocytes represents the percentage of infected erythrocytes. Every 48 h, when the culture was ring stage, it was split to give ~0.5% parasitemia.

The Parasite Erythrocyte Multiplication Rate (PEMR) for each day was calculated by dividing the parasitemia of the ring stage culture by the parasitemia the day before. Custom Python scripts were used to do further analysis. Due to a negative correlation between the starting parasitemia and PEMR of the subsequent cycle, any cycles where the starting parasitemia was >1% were removed. We also manually went through the plots of the PEMR for each cycle by repeat and removed any data sets that had a very high parasitemia (typically greater than 5%), causing subsequent repeats to be very low (the culture had crashed). The measurements were taken over a total of 5–8 invasion cycles, the number of cycles was determined by when blood was available to allow the cultures to be split over the course of the assay using blood from 3 donors to account for potential human sources of technical variation.

## Tight synchronisation

Defrosted lines were sorbitol synchronised in the first cycle when the culture was greater than 0.5% rings and then delayed at room temperature to push the egress window to ~9 am on a Tuesday or Thursday. The above percoll synchronisation was carried out with a 3.5 h invasion window. The infected blood was resuspended at 1.5% HCT in complete media. The flask was delayed at room temperature until 8 pm on Tuesday or if synchronisation was done on a Thursday until 8 pm on Friday to ensure the start of the egress window remained around 9 am on Tuesday and Thursday. This was done repeatedly every Tuesday and Thursday for the duration of the experiments.

## Rapamycin activation and DMSO control treatment of conditional knockouts

All the conditional knockout lines used were made in the B11 background, a clone of 3D7 that expresses DiCre recombinase components [25,48]. When activating the conditional knock-out lines the above tight-synchronisation protocol was used, when the culture was at a parasitemia > 4% and in 1.5 ml of packed infected blood at the start of the Tuesday cycle, the above protocol was followed at the end of which 1 ml of the blood was taken for imaging and the remaining 0.5 ml was diluted with fresh blood to keep the total packed blood at 1.5 ml and kept in the synchronisation protocol. The 1 ml of blood was split into two flasks with 30 ml of media added to each and delayed until 8 pm on Wednesday at which point dimethyl sulfoxide (DMSO) was added to one flask and rapamycin (Sigma-Aldrich) resuspended in DMSO added to the other at a 1:10,000-fold dilution to give a final rapamycin concentration of 10 nM. The flask was returned to the incubator at 37˚C for 14 h to allow rapamycin-induced DiCre-mediated gene deletion to occur. The culture was washed twice with complete media to remove the DMSO or rapamycin before returning to normal culture conditions.

## Protein separation by sodium dodecyl sulphate polyacrylamide gel electrophoresis (SDS PAGE) and Western blot analysis

DMSO (control) and Rapamycin (Rap) treated schizonts, prepared as described above, were solubilised 3:1 in 4X SDS loading buffer (0.2 M Tris-HCl, 277 mM SDS (8% w/v), 6 mM Bromophenol blue, 4.3 M glycerol) and boiled at 95˚C for 5 mins. Proteins in samples were separated, alongside a protein standard molecular weight marker (SeeBlue Plus2, Invitrogen), on 4–15% gradient polyacrylamide gel (Mini-Protein TGX precast protein gels, BioRad) by gel electrophoresis at a constant voltage of 200 V in SDS running buffer (25 mM Tris, 192 mM glycine, 0.1% SDS, pH 8.3).

Following separation, gels were analysed by Western blot, probing for MSP1 (X509, anti-MSP1 p38/42, 1:1000) or SERA5 (pAb Anti-SERA5, 1:2000). The proteins separated in gels were transferred onto nitrocellulose membrane (pore size 0.2 μm, Biorad) via overnight wet transfer (Trans-Blot cell, Biorad) in transfer buffer (25 mM Tris, 192 mM glycine, 20% methanol (v/v), pH 8.3). Nitrocellulose was blocked for 1 h using 5% semi-skimmed milk in PBS 0.05% Tween 20, washed three times in PBS 0.05% Tween 20 and incubated at room temperature with the primary antibody diluted in 3% w/v Bovine Serum Albumin (BSA) for 1 h. The blots were then washed three times in PBS 0.05% Tween 20 and then incubated at room temperature with horseradish peroxidase-conjugated secondary antibody diluted in PBS 0.05% (For X509: Goat-anti-human (Abcam), 1:5000; for pAb anti-SERA5: Goat-anti-rabbit (Abcam), 1:3000) Tween 20 for 1 h. After incubation, blots were washed a further three times for 5 min in PBS 0.05% Tween 20, then horseradish peroxidase substrate (Immobulin, Millipore) was applied and then immediately imaged using a Biorad ChemiDoc imaging system.

## Immuno-fluorescence assay (IFA)

After synchronisation and treatment with Rap or DMSO as described above, parasites were collected at schizogony by using centrifugation on a 70% Percoll cushion. Parasites were then washed in RPMI w/Albumax. Schizonts were placed in PKG inhibitor C2 (4-[7-[(dimethylamino)methyl]-2-(4-fluorphenyl)imidazo[1,2-α]pyridine-3-yl]pyrimidin-2-amine), LifeArc), 1 μM in fresh RPMI w/Albumax, and allowed to mature for 4 h. Schizonts were then smeared onto glass coverslips and allowed to dry completely. If not to be used immediately, dried coverslips were wrapped in tissue paper and stored at -80˚C in water-tight sealed bags with silica beads (desiccant condition). Sealed containers were then warmed to 37˚C for 1 h prior to use. Smears on coverslips were then fixed with 4% formaldehyde (Thermo scientific) in PBS for 1 h and then washed with 0.1% Triton X-100 in PBS for 10 min and then two times in PBS for 5 min. Slides were blocked with 3% w/v BSA (Sigma Aldrich) overnight and then washed three times in PBS. Slides were incubated with primary antibody in 3% w/v BSA (X509, 1/5000) in a humidified chamber at 37˚C for 1 h and then washed three times PBS for 5 min. Incubation with secondary Alexa Fluor antibodies diluted in 3% w/v BSA (1/2000, Goat-anti-human-alexofluor647) was carried out in a humidified chamber at 37˚C for 1 hr, shielded from light and then washed three times in PBS for 5 min. Slides were mounted in Vectashield containing DAPI (Vector laboratories) and images were acquired using a AxioVision 3.1 software on an Axioplan 2 microscope (Zeiss) with a Plan-APOCHROMAT 100×/1.4 oil immersion objective and processed using Fiji.

## Imaging of conditional knock-outs

The conditional knock-out lines were treated as described above then on Friday imaging with the optical tweezers was carried out as described above. Imaging was performed at least three times for each line with the DMSO and rapamycin samples imaged on the same day in the same blood; different blood was used for each experiment, to account for potential human sources of biological variation. A sample of the culture was saved to make Giemsa smears to assess after the invasion window and a pellet of the culture was saved for genotyping on the day of imaging.

## Generation of PfAMA-mNEON

The PfAMA1-mNEON tagged line was constructed by integrating a full-length PfAMA1-mNEON fusion, flanked by 5' and 3' regions of the PfAMA1 gene, into the PfP230P locus. The line also retained episomal copies of the repair template plasmid, meaning that

PfAMA1-mNEON could be expressed from both the PfP230P locus and the episomal plasmid. The 5- and 3' homology regions and guides used here and for the ΔPfP230P described below were based on the design used by [106]. All the sequences used for construction are detailed in, **S4 Table**. A repair template plasmid was constructed using Gibson assemble [107] (NEB Hifi assembly kit). The repair construct was made by joining the PfAMA1 coding region and ~1.5 Kb 5' to the transcription start site amplified from 3D7 genomic DNA, a C-terminal mNeongreen tag and the ~1 Kb region 3' to the end of the PfAMA1 transcript amplified from 3D7 genomic DNA; either side of this assembly were homology regions amplified from PfP230P. The repair template was inserted into the NotI/SacII sites of the pCC1 plasmid (PlasmoGEM Sanger) while removing the negative selection cassette. The guide was inserted into plasmid pDC2-cam-Cas9-U6-yDHODH (PlasmoGEM Sanger), pDC2 contained two bbsI sites in the guide RNA sequence, for each guide a pair of primers were designed which when annealed produce the guide sequence with overhangs matching those produced when pDC2 was digested with bbsI (New England Biolabs, USA), allowing for ligation to occur. pDC2 also contains the cas9 expression sequence.

Ethanol precipitation was used to prepare 60 mg of repair pCC1 plasmid and 30 mg of guide pDC2 plasmid in 60 μl of cytomix (120 mM KCl, 0.15 mM CaCl2, 2 mM EGTA, 5 mM MgCl2, 10 mM K2HPO4, 25 mM HEPES, pH 7.6). Ring stage sorbitol-synchronised culture (parasitemia 4–5% rings) was washed in cytomix. For each transfection 60 μl of DNA and 450 μl of infected erythrocytes were then electroporated at 310 V, Resistance: infinite, 950 μF with a Gene Pulser (BioRad, Watford UK). This was transferred into 6 ml of complete media to give a 4% HCT. The culture was then maintained on WR99210 (WR) (Sigma-Aldrich) until parasites could be seen in Giemsa smears. The selection was done with WR99210, which only selected for the repair template in this instance. The PCC1 repair template plasmid contains the hDHFR gene which gives resistance to otherwise toxic WR99210 (WR) [108]. The line was genotyped, as described in **S4A–S4B Fig** using primers detailed in **S15 Table**.

## Antibody binding assay

Blood was diluted to 4% HCT and washed twice in complete media. For each condition, 100 μl of blood was added to a well of a 96-well plate. The cells were pelleted by centrifugation, the supernatant removed and replaced with 100 μl of the complete medium if indicated containing the primary antibody, detailed **S1 Table**. Control wells were run with just blood or just the secondary antibody. The plate was incubated for 60 min at 37˚C and then washed three times in PBS. Then 100 μl of the second secondary antibody goat-anti-rabbit AlexFluor488-labelled Invitrogen (A32723) at a 1:1000 dilution (2 μg/ml) was added to the appropriate wells. Two washes were then done in PBS and then the samples were diluted to at 0.4% HCT in a round bottom 96-well plate. The sample was then analysed using an Attune NxT acoustic focusing cytometer (Invitrogen) with excitation using a 488 nm laser and detection using a 530/30 filter. 50,000 events were collected per sample. Analysis was done using a custom Python script that used the FlowCal library [109]. Initial gating was done as before to the isolate signal from single erythrocytes. Next histograms for the BL1-A intensity which detects the Alex Fluro plus 488 were normalised to allow comparison of data with different counts. We then averaged the histograms of the repeats.

## Growth inhibition assay

Two-colour assays were set up as described [29]. The day before, the culture was synchronised using sorbitol as described above so that the culture was predominantly late-stage trophozoites at the time of assay set-up with a 1–2% parasitemia and 2% HCT. Uninfected erythrocytes

were stained with 4 μM CellTrace Far Red Cell Proliferation kit (Invitrogen, UK) for 2 h at 37°C whilst rotating and then washed twice in complete media and resuspended at a 2% HCT. 50 μl of the labelled erythrocytes were mixed with 50 μl of the infected culture in the wells of a flat-bottomed 96-well plate. The erythrocytes were pelleted by centrifugation, the supernatant removed and replaced with 100 μl of the complete medium, containing the relevant antibody or inhibitor at the final concentration if indicated. The details of the commercial antibodies used are detailed in **S1 Table**, while the R1 peptide at 98% purity (Sequence: VFAEFLPLFSKFGSRMHILK,GenScript). The first anti-GYPC antibody we tested, BRIC 10, caused the erythrocyte to aggregate, so was not used further. In all assays, wells were set up with no antibodies used as controls to express relative expression, and wells containing only unstained uninfected erythrocytes were included to aid in correctly positioning gates. To obtain neuraminidase-treated erythrocytes, blood was diluted to 20% HCT in RPMI, neuraminidase (Vibrio Cholera N7885-1UN) was added at 66.7 mU/ml (the stock was assumed to be at 1.5 U/ml), and the mixture incubated at 37°C for 1 h on a rotator. After incubation, erythrocytes were washed twice with RPMI and stained with CellTrace Far Red as above.

Plates were incubated for 20 h under *P. falciparum* culture conditions described above. After incubation, infected erythrocytes were stained with the fluorescent DNA dye SYBR Green I (Invitrogen, Paisley, UK) at a 1:5000 dilution for 1 h at 37°C. The sample was then fixed by suspending in 2% paraformaldehyde and 0.2% glutaraldehyde in PBS and incubating at 4°C for 30 min. The sample was then resuspended at 0.4% HCT in PBS. The samples were then analysed either with a BD Fortessa flow cytometer (BD Biosciences, Oxford, UK) running BD FACS Diva software (BD Biosciences, Oxford, UK) or an Attune NxT acoustic focusing cytometer (Invitrogen). On the BD Fortessa, SYBR Green I was excited with a 488 nm blue laser and detected by a 530/30 filter and CellTrace Far Red was excited by a 640 nm red and detected by a 670/14 filter. On the Attune NxT, SYBR Green I was excited with a 488 nm laser and detected by a 530/30 filter, and Cell Trace Far Red was excited by a 637 nm laser and detected by a 670/14 filter. 50,000 events were collected per sample. Analysis was performed using FlowJo (Tree Star, Ashland, Oregon). Initial gating was done as before to isolate the signal from a single erythrocyte. The percentage of the CellTrace Far Red labelled erythrocytes that had been infected (ensuring only erythrocytes infected after the assay was started are measured) was determined using the plots of the blue laser (SYBR Green) vs red laser (Cell trace Far Red). Unless indicated otherwise, all experiments were carried out in triplicate.

When testing the R1 peptide, even at the highest concentration we still detected some invasion, we explored whether the residual level of parasites identified by flow cytometry represented real invasion events, or merozoites stuck on the outside of erythrocytes, which could happen after blocking PfAMA1 function, and which cannot be distinguished between using flow cytometry. Using fluorescence microscopy with both the untreated control and the sample treated with 40 μM R1 peptide (the highest concentration tested), we saw both rings (star-shaped DNA signals) in erythrocytes as well as merozoites that appeared to be stuck on the outside. This indicates that some of the parasitemia signal present with the 40 μM R1 peptide after invasion is rings; we did not, however, maintain the culture to see if the parasites could develop further.

### Generation of CRISPR knock-out lines

Knockouts were generated using a CRISPR/Cas9 approach. Construct design was done in Benchling and gene sequences were obtained from the PlasmoDB website (https://plasmodb. org/plasmo/app). Details of all the sequences used for each line are summarised in **S3 Table**.

Each knock-out was constructed by replacing a region of the gene with a selectable marker. Guide sequences were selected in the region to be replaced, **Fig 3A**, using the CRISPR tool on Benchling. The guide plasmids for the knock-out line construction were made in the same manner as described above for the PfAMA1-mNeonGreen line. Repair constructs were created using the pCC1 plasmid (PlasmoGEM Sanger) backbone, in which homology regions (500–800 bp) targeting the region of interest were assembled on either side of the human dihydrofolate reductase (hDHFR) positive selection maker and a barcode sequence unique to the knock-out; the plasmid also contained a negative selection gene for cytosine deaminase to allow the plasmid to be removed from the edited parasites [69]. The homology regions were amplified from NF54 DNA using primers specific to the homology region with 20 bp overhangs added to allow annealing to the neighbouring fragments. The hDHFR gene was amplified from the pCC1 plasmid. The plasmid backbone was made by digesting pCC1 with SpeI and EcoRI then gel purifying the cut band. The repair template was assembled and integrated into pCC1 using Gibson assembly [107] (NEB Hifi assembly kit). Colony PCR was used to check the plasmids had the correct inserts (F-93/R-91) and (F-92/R-94), sequences in **S2 Table**.

Transfection was done as described above. Once transfected the parasites were cultured in the presence of 2.5 nM WR99210 (Jacobus Pharmaceuticals) until the parasites could be seen in Giemsa smears. For the lines made as part of the pipeline [110] (ΔPfEBA140, ΔPfEBA181 and ΔPfRH4) the lines were grown from stocks frozen after transfection and positively selected by growth on WR99210 then treated like the other lines. The presence of WR99210 selects for both edited parasites where the repair template containing the hDHFR gene had been integrated into the target locus and wild-type parasites carrying the pCC1 plasmid as an episome. Negative selection was therefore carried out by culturing the parasites in the presence of 5-fluorocytosine (5-FC, Sigma-Aldrich, UK) at 48 mM and 2.5 nM WR99210 for a week. The cytosine deaminase gene on the pCC1 plasmids makes the 5-FC lethal [111], so any parasites still carrying the plasmid were killed. This ensures that the only parasites remaining are the ones with the hDHFR-positive selection gene integrated into the targeted genome region.

Finally, all the lines were cloned by limiting dilution using a plaque cloning assay as described by [112]. Wells containing a single colony were inspected using an EVOS microscope (Leica, Germany), 4X objective. Finally, PCR (CloneAmp HiFi PCR Premix, Takara-Bio Europe) genotyping was done with primers designed for each knock-out line that allowed for confirming either the presence of the wild-type (F-gt5/R-gt5WT and F- gt3WT/R-gt3) or presence of the knock-out gene (F-gt5/R-91 and F-92/R-gt3), as shown in **S8A Fig**. Sequence given in **S2 Table**.

We successfully made cloned knock-outs of PfEBA140 (PlasmoDB ID PF3D7_1301600), PfEBA181 (PlasmoDB ID PF3D7_0102500), PfRH1 (PlasmoDB ID PF3D7_0402300), PfRH4 (PlasmoDB ID Pf3D7_0424200), PfRH2a (PlasmoDB ID PF3D7_1335400), PfP230P (PlasmoDB ID PF3D7_0208900) and Pfs25 (PlasmoDB ID PF3D7_1031000). We were unable to make ΔPfRH2b (PlasmoDB ID PF3D7_1335300), 4 attempts were made at transfecting using 3 different guides. On several of these occasions, other transfections were done in parallel, resulting in successful edited parasites. The same guides and repair construct worked on the first attempt when the transfection was done into the 3D7 line and correct editing was shown by genotyping indicating the constructs worked. The efficiency of disrupting PfRH2b in NF54 seems very low, whereas ΔPfRh2b lines have been made in 3D7 multiple times previously [52,69]. Given all the other knockouts were made on the NF54 background, complicating comparisons, we did not use the 3D7 ΔPfRH2b for further studies.

## Quantification of gene expression with RT-PCR

Protocol available DOI: dx.doi.org/10.17504/protocols.io.q26g7peo9gwz/v1. The qPCR set-up was guided by the protocol described [77]. All the lines were synchronised in parallel with the tight synchronisation protocol described above, using a 150 ml culture at a 1% HCT. On a Thursday morning, the late-stage schizonts were isolated with Percoll as described above, a third was left to reinvade and two-thirds of the schizonts were added to 40 ml media with PKG inhibitor Compound 2 ((4-[7-[(dimethylamino)methyl]-2-(4-fluorophenyl)imidazo [1,2-α] pyridine-3-yl]pyr- imidin-2-amine)) at 1 μM at 37˚C for 3.5 h. Compound 2 arrests schizonts ~15 min prior to egress [85], ensuring the samples were all comparable in their developmental stage. After C2 treatment, the schizonts were then pelleted by centrifugation, washed in 5 ml ice-cold PBS, and resuspended in 1 ml ice-cold 0.01% saponin in PBS, left on ice for 10 min then pelleted again and resuspended in 1 ml of 0.01% Saponin, pelleted and resuspended in 200 μl PBS (100 μl if the pellet was small). 2000 μl of TRIzol LS reagent (Ambion ref 10296010) was added and heated to 37˚C for 5 min. We collected samples for all the lines in parallel over 4 weeks in different blood each week, collecting one sample a week with two invasion cycles in between.

1000 μl of each sample in Trizol was then added to Phasemaker tubes (Invitogen A33248) and left for 5 min, then 200 μl of chloroform was added, left for 3 mins at room temperature and then centrifuged at 12,000 g for 5 min at 4˚C. The RNA was then extracted from the aqueous phase using the RNA Clean and concentrator kit (Zymo R1015) following the manufacturer's instructions and using the on-column DNase I treatment. cDNA synthesis was performed using 1 μg of RNA per sample with the superscript IV VILO MM with ezDNase enzyme (ThermoFisher 11756050) following the manufacturer's instructions alongside a no reverse transcription (RT) control. For each qPCR reaction, the SYBR Power SYBR Green PCR Master Mix (Fisher scientific 4367659) was used with a 10 μl volume and a 650 nM primer concentration; the sequence for primers used is given in **S3 Table**. For every pair of primers, a set of standards of genomic NF54 DNA was run at (10, 2, 0.4 and 0.08 ng/μl) on every plate. We ran a minimum of triplicate wells for the samples and the standards, then one for the no-RT control and water controls. Samples were pipetted into the 384-well plates using an Opentrons OT2 pipetting robot with an Opentrons Gen 2 p20 pipette to minimise technical variation, keeping the 384-well plate cooled to 10˚C. Each set of samples was run over two plates, with all lines on each plate and primers for 7 genes with Actin I (housekeeping reference) repeated on both plates. The plates were run using a BioRad 384 well RT PCR machine (CFX384) at 95˚C for 10 min, followed by 40 cycles of 95˚C for 15 s, 55˚C for 20 s, 60˚C for 60 s.

Analysis was done using a custom Python script. Any outliers were eliminated by removing any points of the triplicate repeat that were more than 0.3 Cq away from the mean of the data points (the measurements for 47 wells out of the 1470 measured for the cDNA of the different lines were removed representing 3.2% of the measurements). For each set of primers, we performed a linear fit to the measurements of the standards. The Cq values were interpolated based on the standard curve. We confirmed visually that there was a good separation between the RT samples and the no-RT control. Each gene's expression levels were then normalised to the housekeeping gene Actin I in the given sample. The gene expression of each knock-out line is expressed as a relative fold change compared to the wild-type NF54 line, ΔPfP230P or ΔPfs25. To determine which changes were significant we performed a one sample t-test with the null hypothesis that the true mean of the population equals 1 at a 95% level of significance, which would represent no change in gene expression relative to the corresponding reference line.

## Statistics

Error bars used throughout show the standard error of the mean or the standard deviation as indicated in the legends of the figures. The Anderson-Darling test has been used to assess if the detachment force data sets are normally distributed [113]. The test assesses how well a set of data fits a normal distribution. The null hypothesis is that the data are normally distributed. The test gives a P value of how likely the null hypothesis is true. P = 0.05 was used to determine if the dataset fulfilled the null hypothesis. If both data sets were normally distributed, as assessed with the Anderson-Darling test, the significance of the difference between them was assessed with a t-test at a 5% significance level. The test assesses the null hypothesis that the two data sets being compared are normally distributed and come from independent random samples with equal means and equal but unknown variances [114]. If one or both data sets were not normally distributed, assessed with the Anderson-Darling test, then the difference's significance was assessed with a Wilcoxon rank test at a 5% significance level. The non-parametric test assesses the null hypothesis that the two data sets are from a continuous distribution with equal medians. The test assumes that the two samples are independent [114].

## Supporting information

**S1 Video. This video shows an example of measurements made from one egress of the wild-type strain NF54.** In all examples, one erythrocyte is attached to the glass slide, and the other is stretched with an optical trap. The video indicates what was counted at an erythrocyte-merozoite-erythrocyte position and an attachment (from which a detachment force was measured).
(MP4)

**S2 Video. This video shows examples of detachments that occurred when both erythrocytes were held by an optical trap for wild-type strain NF54.**
(MP4)

**S3 Video. This video shows an abnormal egress measured for rapamycin-activated cΔPfMSP1 and a detachment measured for one of the released merozoites.** Then, there is a further example of a clumpy egress.
(MP4)

**S1 Fig. Exploration of the relationship between detachment force and egress time.** As merozoites are known to lose invasive capacity over time, it is possible that egressed merozoites lose attachment potential/strength over time. All measurements were taken within 180 s of egress, but we nevertheless compared the detachment force and the time post-egress that each measurement was made. No correlation was found between detachment and time post-egress. **(a-e)** Plot of the time post egress that the detachment force was measured against the measured detachment force, with strain measured above each plot. **(c-f)** contains data gathered from the inactivated DMSO-treated controls for the three conditional knock-out lines discussed in **Fig 2**, all made in the 3D7 background. cΔPfAMA1, cΔPfMSP1 and cΔPfGAP45 showed no correlation between time and force (correlation coefficient < 0.1), NF54 showed a very weak positive correlation, correlation coefficient = 0.15 and 3D7 showed a weak negative correlation, correlation coefficient = -0.59, although for this strain there were only 23 points plotted. This indicates that there is no relationship between time post-egress and detachment force. **(f)** For some merozoites, multiple attachments were made using the same or different erythrocytes. In these cases, the detachment force vs the time post egress for each measurement has been plotted, with lines connecting the attachments of the same merozoite. Only merozoites

with more than 2 attachments measured were plotted. Each merozoite is represented by a different style line and marker. This shows that detachment forces do not consistently drop with multiple attachments even after up to 9 measured detachments.
(TIFF)

**S2 Fig. Comparing detachment forces measured using the same merozoite.** During the optical tweezer manipulation sometimes one erythrocyte becomes attached to the glass slide and only one erythrocyte is stretched by the optical trap, **S2 Video**, whereas in other cases both erythrocytes are held by optical traps and hence both become stretched, meaning that two forces can be measured for these detachments. We, therefore, compared detachment forces from both erythrocytes and found a positive but weak correlation. This is likely due to biological variation between erythrocytes, as is that we assume that the stiffness constant of all erythrocytes is constant (20 pN/μm); however, there is known to be at least 20% variation between erythrocytes within a single donor [37]. **(a)** Cartoon shows the two ways that the detachment forces are measured. When a merozoite is attached to two erythrocytes either one erythrocyte is attached to the slide and one trap is used to stretch the other erythrocyte, from which the force is measured, or neither erythrocyte is attached to the slide, meaning traps are used to hold both erythrocytes. In this latter case, both erythrocytes stretch so two measurements are recorded for the detachment. In the rest of the paper in the second case, one force is randomly selected and included, and the other is ignored. Error bar = 5 μm. **(b and c)** NF54 wild-type data was originally discussed in **Fig 1C**. For the conditional knockouts cΔPfMSP1, cΔPfAMA1 and cΔPfGAP45 the inactivated DMSO treated controls are plotted, discussed initially in **Fig 2**. The conditional knockouts are made in the 3D7 background. **(b)** The force is measured from erythrocytes x and y when two erythrocytes are stretched while attached to the same merozoite. The assignment of a and y to the erythrocytes is random. The grey line shows the fit of the scatter plot. The green line shows the expected fit if x and y were equal. **(c)** Box plots showing merozoite-erythrocyte detachment force. The central bold line shows the median, with the top and bottom of the box at the 25th and 75th percentiles and the whiskers showing the total range of the data. Plotted are all the forces measured for erythrocyte x, and erythrocyte y, the mean of detachment force of x and y for a detachment and the single forces that were measured when only one erythrocyte was stretched. **(c)** Data sets that were significantly different (t-test) have a line between them with stars showing the level of significance * = ≤0.05, ** = ≤0.01, and *** = ≤0.001.
(TIFF)

**S3 Fig. Generation of a PfAMA1-mNEON tagged line and its use to establish whether detachment consistently occurs from one merozoite end.** Because merozoites are polar cells it is possible there is a bias to the side that detachment occurs from. To explore this, we attempted to label the apex of the merozoite to see whether detachment predominantly occurs from one end. **(a)** A line was constructed in the 3D7 background where an additional copy of Apical Membrane Antigen 1 (PfAMA1) C-terminally tagged with the fluorophore mNEON (PfAMA1-mNeon) was integrated into the PfP230P landing site. The diagram represents the genetic structure of the PfAMA1-mNEON line. The top shows the homology regions in the PfP230P gene before PfAMA1-mNEON was integrated. The bottom shows the same region after editing. The sequences for the endogenous PfAMA1 5'UTR and PfAMA1 coding region were linked to the fluorophore mNEONgreen followed by the 3'UTR for the endogenous PfAMA1. The approximate positions of the primers (shown with arrows) to genotype the lines and the letters in between a pair of primers show the reaction ID. (b) Genotyping PCR was used to confirm that the correct edit had been made. The gel for the PCRs run on both the PfAMA1-mNEON line and the unedited parent line 3D7 for reference. The primers used for

each reaction are summarised underneath and the sequences are given in S2 Table. Reactions A and B confirm that PfAMA1-mNEON was correctly integrated into the PfP230P locus. C and D confirm the absence of any unedited parasites. Reactions E and F confirm the presence of the homology repair plasmid. PfAMA1-mNeon can also be expressed from the plasmid. (c-d) Show images taken with the same microscope used to carry out all the optical tweezer measurements in this paper. In schizonts PfAMA1 localises to the micronemes and is released at egress, meaning it is initially primarily apically localised, but then diffuses over the merozoite surface over time [100,115]. The left image shows brightfield, the centre image shows the signal from the PfAMA1-mNEON and the right image shows the overlay of the other two. Error bar = 5 µm. (c) Shows that the signal from PfAMA1-mNEON can be seen in schizonts before egress as distinct foci. (d) Shows the frame before the detachment of an attached merozoite. PfAMA1-mNeon signal can be detected in the merozoite but the signal is too weak/defused to determine the orientation of the merozoite as hoped. Eight attachments were observed, and the signal was not bright/clear enough in any of them. (e) We assessed the NF54 data set to see whether the merozoite was predominantly detached from the first or second erythrocyte it had attached to, reasoning that the erythrocyte that was attached first is more likely to be attached to the apex. The box plot shows the detachment force of merozoite-erythrocyte attachment. The central bold line shows the median, with the top and bottom of the box at the 25[th] and 75[th] percentiles and the whiskers showing the total range of the data. For 62 attachments for the NF54 strain where we could clearly see which erythrocyte attachment occurred first, 68% of the time detachment occurred from the second erythrocyte the merozoite attached to, suggesting some level of preferential detachment. However, the mean detachment force was not affected by whether the detachment was from the first or second erythrocyte the merozoite attached to; from the first erythrocyte the mean was 25 ± 2 pN and from the second 23 ± 3 pN, (p = 0.52 t-test).
(TIFF)

**S4 Fig. Cas 9 mediated gene editing of PF3D7_0930300 locus to allow conditional knock out of PfMSP1 (cΔPfMSP1).** Further characterisation of this line will be published elsewhere. **(a)** Mutagenesis strategy. The integration construct was designed to integrate re-codonised PfMSP1 (rc. PfMSP1, dark grey) floxed with head-to-head oriented lox66/77 (black, triangles) into the endogenous Pfmsp1 locus (g. PfMSP1) of the parental line A7 which is based on a 3D7 background. Integration is guided by the 5' and 3' homology sequence (white, HR5 and HR3). The CRISPR Cas9 cassette co-transfected with the integration construct encodes Cas9 and a guide sgRNA. This ensures a targeted double-stranded break in the endogenous PfMSP1 sequence (PF3D7_0930300), that allows the insertion of the integration construct. The hdhfr gene (light grey) confers resistance to the antifolate WR99210, allowing the selection of transfected parasites. Treatment with rapamycin (rap) activates DiCre, which mediates inversion of the floxed sequence, introducing premature stop codons. Only a ~16 kDa truncated form of PfMSP1 is encoded by the modified locus, and this was not detectable in cells and likely not expressed. The line produced 3D7MSP1KO:lox66/lox71rev is referred to here as cΔPfMSP1. Coloured arrows show the primer binding position. **(b)** Genotyping gel of PCRs were run to confirm editing. Reaction A (blue arrow, primer-229 and red arrow primer–230) prime off in the endogenous PfMSP1-D sequence demonstrated successful integration at the expected locus (expected product for parental line A7, +/- Rap 2000 bp; expected product after integration, cΔPfMSP1 +/- Rap, 2282 bp). To monitor sequence inversion, an oligo was designed to prime off the integrated inverted recodonised (rc.) sequence (yellow arrow, primer 228)) when paired with the endogenous N-terminal forward primer (red arrow primer 229). Upon the addition of Rap and sequence inversion, a 2039 bp product is expected. No product is expected

for DMSO-control treated cΔPfMSP1 parasites and for the parent line (A7) +/- Rap. The sequence of the primers used, are given in **S2 Table**. **(c)** Evidence of loss of PfMSP1 expression upon activation of gene disruption. SDS-PAGE and Western blot analysis of DMSO (control) or Rap-treated cΔPfMSP1, parasites were harvested from compound 2 (C2) arrested schizonts. The samples were probed with anti-PfMSP1 p38/42 (mAb X509) which showed a reduced expression of PfMSP1 in Rap-treated parasites. The samples were also probed with anti-SERA5 as a control. Lader used is SeeBlue Plus2 from Invitrogen. **(d)** Indirect fluorescent antibody test (IFA) of C2 arrested cΔPfMSP1 schizonts demonstrating loss of PfMSP1 expression. This line constitutively expresses EXP2 labelled with mNeon (emission 488nm), allowing visualisation of the PVM. DAPI is used as a nuclear stain. Both Rap-treated and untreated (DMSO) schizonts were probed with mAb X509 (binds MSP1 p38/42). Loss of signal for Rap-treated schizonts demonstrates loss of PfMSP1 expression, the assessment of 423 schizonts showed a 94.32% excision rate. Scale bar 20 μm.
(TIFF)

**S5 Fig. Validation of conditional knock-out lines used of optical tweezer experiments (a, c-d)** Rapamycin-induced deletion was validated for all three-conditional knock-out lines tested in **Fig 2** at the same time as imaging was carried out. Samples of rapamycin-treated (Rap) and DMSO-control (the solvent in which rapamycin is dissolved) treated parasites were collected and washed in PBS, and genotyping PCR was used to confirm gene excision. All gels are for genotyping PCRs, indicated are the primers used, sequences are given in **S2 Table** as well as the expected sizes for the bands in the modified locus (locus with integrated loxP sites) and the excised locus that should be present after rapamycin treatment. The numbers indicate the repeat number. The ladder is NEB 1 kb DNA Ladder (N0552). Tables show a summary of the parasitimia assessed at the beginning of the next cycle after DMSO/Rap addition by counting the number of rings in Giemsa smears. **(a)** Measurements were collected for the cΔPfMSP1 line. **(b)** For cΔPfMSP1 egresses were assessed during the optical tweezer assays and classified as normal, clumpy or unsure (for egresses that we were not sure if they were clumpy as they were hard to distinguish). The table summarises what fraction of the egresses fell into each class. Box plot showing the detachment force of merozoite-erythrocyte attachment. The central bold line shows the median, with the top and bottom of the box at the 25th and 75th percentiles and the whiskers showing the total range of the data. The bar chart shows the frequency of positioned cells that lead to attachment. The distributions were compared with a t test, and there were no significant differences at a 5% level of significance. Error bars show the SEM. **(c)** Measurements were collected for the cΔPfAMA1 line. **(d)** Measurements were collected for the cΔPfGAP45 line.
(TIFF)

**S6 Fig. Validation of inhibitors and antibodies used to disrupt invasion. (a)** Validation of antibody binding to erythrocytes. Primary antibodies used were anti-CR1 (Ab25 –E11), anti-Basigin (AB119114 MEM-M6/6), anti-GYPA (BRIC 256), anti-GYPC (BRIC 4) and control antibody (which should not bind erythrocytes) anti-PKHG1 (ab121979). Erythrocytes were incubated with each of the primary antibodies, and then a secondary anti-mouse Alexa Fluro plus 488 conjugated antibody was used to detect binding in flow cytometry. Data was collected for three independently labelled wells for blood from two donors. The area under each histogram was normalised to have the same area and then averaged for each condition. The erythrocytes (unlabelled), anti-mouse secondary antibody and the control anti-PKHG1 all showed negligible binding, whereas the histograms for the anti-CR1, anti-Basigin, anti-GYPA and anti-GYPC all showed increased fluorescence intensity relative to controls, indicating all primary antibodies used were able to bind to the erythrocytes. **(b-d)** Growth inhibition assay

done using wild-type strain NF54. When no treatment was present, a 100% invasion efficiency was set as the percentage of invaded erythrocytes. Error bars show the standard error of the mean (SEM). **(b)** Investigates the effect of antibodies, anti-GYPC data based on two repeats. **(c)** Investigates the effect of the R1 peptide inhibitor. **(d)** Investigating the effect of invasion into blood treated with 66.7 mU/ml neuraminidase, mean invasion efficiency was 60 ± 4%, based on two repeats. **(e)** Anti-CR1 E11 binds erythrocytes (a) but has no detectable effect on invasion, even at the highest concentration of 10 μg/ml (b). However, we still tested for an effect on attachment with the optical tweezers. Only one biological repeat was done for NF54 +anti-CR1 antibody. Shown is a box plot showing the detachment force of merozoite-erythrocyte attachment. Mean detachment force NF54 24 ± 2 pN; NF54+anti-CR1 26 ± 4 pN, not significant compared to NF54 (t-test p = 0.66). The central bold line shows the median, with the top and bottom of the box at the 25th and 75th percentiles and the whiskers showing the total range of the data. **(f)** The bar chart shows the frequency of erythrocyte-merozoite-erythrocyte positions that lead to attachment of the merozoite to both erythrocytes. Attachment frequency per egress for NF54 mean 19 ± 3% from 552 positions; NF54+anti-CR1 mean 22 ± 7%, number positions = 90. The distributions were compared with a Wilcoxon rank test as the NF54 data set was not normally distributed (Anderson-Darling test), and there was no significant difference (p = 0.14). Error bars show the SEM.
(TIFF)

**S7 Fig. Correlation between invasion efficiency and attachment behaviour measured with optical tweezers.** Correlation of the invasion efficiency relative to wild-type line NF54 (based on measurements in **Figs 1E, S5B and S5C**) to the detachment force (**Figs 1C and 2C**) and attachment frequency (**Figs 1D and 2C**) for the wild-type line NF54, 3D7 and NF54 in the presence of 10 μg/ml of antibodies anti-GYPC (BRIC 4), anti-GYPA (BRIC 256) or anti-Basigin (MEM-M6/6); or with neuraminidase treatment 66.7 mU/ml erythrocytes or 20 μM R1 peptide that inhibits PfAMA1-PfRON2 binding. The grey line and text show the correlation of all the data points. The red line shows the correlation of NF54, Anti-GYPA, neuraminidase, anti-GYPC and 3D7, excluding the inhibitors that target late-in invasion (R1 and anti-basigin) and so likely affect invasion after attachment. Error bars show the standard error of the mean (SEM).
(TIFF)

**S8 Fig. Genotyping PfEBA and PfRH knock-out lines. (a)** Schematic to represent the genetic change made in the knock-out lines. The red arrows show the approximate positions of the primers used to genotype the lines and the letters in between a pair of primers refer to the reactions shown in the panels below. **(b)** Show gels for genotyping PCRs; details of the primers used are shown to the side as well as the expected sizes for the bands in the unedited locus and the locus after editing; primer sequences are given in **S2 Table**. The only knock-out with a different structure is ΔPfEBA175, as the selectable marker was introduced in the opposite direction, so the primers that bind to the selectable marker were used in the opposite pairs: E– forward primer in front of HR5 and 92 and F– 91 and the reverse primer next to HR3. For each gene, the primers were tested with samples of the knock-out clones used for phenotyping along with a sample of NF54 as a control for what the bands for the unedited locus look like. The ladder is NEB 1 kb DNA Ladder (N0552).
(TIFF)

**S9 Fig. Expression profiles of invasion genes in PfEBA and PfRH knock-out lines assessed by qPCR.** Error bars show the standard deviation between repeats. **(a)** Comparison of gene expression in samples collected from tightly synchronised NF54 line both before C2 treatment

(green) and 3.5 h after C2 treatment (blue). Data represents the mean from three samples collected on different days (biological replicates), and expression is plotted as the fold change relative to the PfActin I housekeeping gene in the sample. Gene expression was very similar before and after compound 2 treatment. A 3.5 h C2 treatment was used for the collection of all other samples. **(b)** Shows the expression profiles for invasion genes in the knock-out lines (the gene deleted in each line is indicated above the graphs), Cq values were interpolated to a standard curve of genomic DNA, with expression in each sample normalised to the housekeeping gene PfActin I and presented as fold-change relative to the expression of those same genes in the control NF54ΔPfP230P line. Four samples for each line were collected (biological replicates collected with invasion into different blood), the samples for all lines were collected in parallel. For each sample triplicate wells were run (technical replicates). NF54 showed no significant differences in any gene tested relative to ΔPfP230P. Across all the lines tested, there was no significant change in PfAMA1 and PfRH5 expression, as expected as they are both essential genes and are not known to be variably expressed; this also confirms that the samples were all consistently synchronised. The control line NF54ΔPfs25 showed significant 14-fold downregulation of PfEBA181 relative to NF54ΔPfP230P (t-test p = 0.0012; interestingly, this was the line that showed significantly ~30% higher invasion rates compared to NF54 and NF54ΔPfP230P, [Fig 3A]). The NF54ΔPfRH4 line showed an even larger 179-fold downregulation of PfEBA181 expression (t-test p = ≤0.0000) (the Cq values were close to the no reverse transcriptase controls for all samples tested, meaning an almost absence of PfEBA181 expression in these lines). PfEBA140 was also notably variable between lines, but the difference was only significant in NF54ΔPfEBA181 where there was 2.8-fold downregulation (t-test p = 0.044). In NF54ΔPfRH2a there was a small but significant 1.4-fold increase in the expression of PfRH2b (t-test p = 0.034). Expression of PfRH3 was significantly lower (t-test p = 0.021) in NF54ΔPfEBA140. In addition, PfRH1, whilst only significantly different in NF54ΔPfRH1 (t-test p = 0.0036), showed high variability in expression, as demonstrated by the large standard deviation in its expression change for several of the lines. Stars indicate a change in gene expression that was significant at greater than a 5% level of significance (t-test). (TIFF)

**S10 Fig. Testing the attachment phenotype of Dd2ΔPfEBA175.** The antibody anti-GYPA (BRIC 256) was used at 10 μg/ml. **(a)** The bar chart shows the frequency of erythrocyte-merozoite-erythrocyte positions that lead to attachment of the merozoite to both erythrocytes. **(b)** The box plots show the measured detachment forces. The central bold line shows the median, with the top and bottom of the box at the 25th and 75th percentiles and the whiskers showing the total range of the data. Error bars show the SEM. (TIFF)

**S1 Table. Details of antibodies used in this study.** (DOCX)

**S2 Table. Details of the primers used in this study.** (DOCX)

**S3 Table. Details of sequences used to construct the knock-out lines used in this paper.** Table summarising the sequences used in constructing the liens made for this paper. This includes the two guide sequences used (guide 1 and guide 2). The start and end sequences of the homology regions that were used are listed which are the sequences used as the overhand in the Gibson assembly (Hr5_F, Hr5_R, Hr3_F and Hr3_R). Finally, the sequences are given for the genotyping primers used to confirm correct integration in the knock-out lines (Gt5,

Gt5WT, Gt3 and GT3WT).
(DOCX)

**S4 Table. List of primers used in qPCR.** F—forward, R—reverse. Ref—reference sequence originally published in. Length—Length of PCR.
(DOCX)

**S1 Data. Data on multiple forces measured for the same merozoite, data plotted in S1 Fig.**
(XLSX)

**S2 Data. Data on the time post egress that detachment forces were measured, data plotted in S1 Fig.**
(XLSX)

**S3 Data. Data for forces measured when one erythrocyte was stretched and when two erythrocytes were stretched, data plotted in S2 Fig.**
(XLSX)

**S4 Data. Data for the forces for wild-type NF54 data grouped by whether the merozoite remained attached to the first or second erythrocyte it initially attached to; data plotted in S3 Fig.**
(XLSX)

**S5 Data. Data for the GIA assays run on antibodies and inhibitors using wild-type strain NF54, data plotted in S6 Fig.**
(XLSX)

**S6 Data. Data for the optical tweezer attachment assays presented in the paper.**
(XLSX)

**S7 Data. Data for the growth assay results presented in the paper.**
(XLSX)

## Acknowledgments

This research was supported by the CIMR Flow Cytometry Core Facility and we wish to thank Reiner Schulte in particular for advice and support in flow cytometry. We would also like to thank Theresa Feltwell, Nadia Cross and the CIMR and Physics of Medicine Department support staff for their invaluable logistical support. We would like to thank Mike Blackman (Francis Crick Institute) for advice and input throughout, Mike Blackman and Helen Saibil (Birkbeck) for the gift of the cΔPfMSP1 parasites the construction of which was partly funded by the MRC Grant (https://www.ukri.org/councils/mrc/) awarded to them 'Membrane and host cytoskeleton reorganization during malaria parasite egress from erythrocytes'. (MR/P010288/1), and Fiona Hackett (Francis Crick Institute) for technical assistance and advice.

## Author Contributions

**Conceptualization:** Emma Kals, Viola Introini, Pietro Cicuta, Julian C. Rayner.

**Data curation:** Emma Kals, Morten Kals.

**Formal analysis:** Emma Kals, Morten Kals.

**Funding acquisition:** Emma Kals, Pietro Cicuta, Julian C. Rayner.

**Investigation:** Emma Kals, Rebecca A. Lees.

**Methodology:** Emma Kals, Eleanor Silvester, Trishant Umrekar.

**Resources:** Rebecca A. Lees, Alison Kemp, Christine R. Collins, Jurij Kotar.

**Software:** Morten Kals.

**Supervision:** Viola Introini, Eleanor Silvester, Pietro Cicuta, Julian C. Rayner.

**Visualization:** Emma Kals, Morten Kals.

**Writing – original draft:** Emma Kals.

**Writing – review & editing:** Morten Kals, Rebecca A. Lees, Viola Introini, Alison Kemp, Eleanor Silvester, Jurij Kotar, Pietro Cicuta, Julian C. Rayner.

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
