## [Decision Letter · Decision Letter 0]

12 Mar 2024

Dear Ms Kals,

Thank you very much for submitting your manuscript "Optical tweezers reveal that PfEBA and PfRH ligands, not PfMSP1, play a central role in Plasmodium-falciparum merozoite-erythrocyte attachment" for consideration at PLOS Pathogens. As with all papers reviewed by the journal, your manuscript was reviewed by members of the editorial board and by several independent reviewers. In light of the reviews (below this email), we would like to invite the resubmission of a significantly-revised version that takes into account the reviewers' comments.

As you can see below, all reviewers were very positive and thought that this manuscript reports innovative, interesting and highly relevant data to the field of invasion biology. While no further experiments are required, there was an important point raised about MSP1 by one reviewer that either needs altering the conclusions or more experiments. Similarly, a EBA175 knock-out is mentioned but not used. Please follow the suggestion of the reviewer who raised that point to either remove the mention of the knock out or include the data if there is any.

In addition, all reviewers indicated that the manuscript contains many small errors and that the presentation of the paper needs significant changes to improve readabiility and accuracy (e.g. "The study is well conducted and I don't have any experimental suggestions but the written part of the paper should be improved by condensing the text " or “Although the overall data in this manuscript is important, the manuscript in its current form is full of errors/inaccuracies and needs a significant and careful re-assessment by the authors.”). All reviewers made excellent suggestions how to amend this that should be followed.

We cannot make any decision about publication until we have seen the revised manuscript and your response to the reviewers' comments. Your revised manuscript is also likely to be sent to reviewers for further evaluation.

Sincerely,

Tobias Spielmann

Academic Editor

PLOS Pathogens

Jeffrey Dvorin

Section Editor

PLOS Pathogens

Michael Malim

Editor-in-Chief

PLOS Pathogens

orcid.org/0000-0002-7699-2064

Reviewer's Responses to Questions

**Part I - Summary**

Reviewer #1: This paper reports on a technically highly challenging study investigating the molecular mechanisms governing red blood cell invasion by the major malaria causing parasite. It’s innovative and provides important and interesting insights. Using a recently established optical tweezer assay in combination with inhibitors and genetically modified parasite lines the authors probe the entry of P. falciparum merozoite into red cells. The probe the attachment frequency and strengths of merozoites. Specifically, they provide evidence that not MSP1 but RH4 is important for strong attachment of the parasite to the red blood cell and that GYPA on the red cell surface also plays a key role.

Reviewer #2: Optical tweezers were used to manoeuvre human RBCs to pick up newly released P. falciparum merozoites and to bring the parasites to other RBCs. Merozoites adhere to RBCs prior to invading them and so by using the tweezers to ‘pull apart’ the RBC-merozoite-RBC sandwiches, the strength of merozoite binding could be inferred by measuring the stretching deformation of the RBCs which have well-established elastic properties. The biggest surprise of the work was that disruption of gene encoding the major merozoite surface protein MSP1, produced no reduction in binding strength suggesting MSP1 is not involved in RBC attachment. This runs counter to many years of work although recent genetic knockout experiments indicate MSP1 is probably more important for the egress of merozoites during schizont rupture. Further work with the knockouts of merozoite genes for EBA and RH proteins as well as treatments which block their corresponding RBC receptors indicates that EBA175 and RH4 might be the most important merozoite ligands for primary RBC attachment.

Reviewer #3: This manuscript by Kals and colleagues titled “Optical tweezers reveal that PfEBA and PfRH ligands, not PfMSP1, play a central role in Plasmodium falciparum merozoite-erythrocyte attachment” dissects in detail binding strength, attachment frequency and invasion capacity of Pf merozoites. The authors succeed in a clear dissection of the contribution of individual ligand-receptor pairs for RBC invasion. Whereas previously published studies had identified the role of individual ligand receptor pairs by analysing real time imaging of the invasion process, quantitation of this data was difficult. Still, the field had accepted that EBAs and RH parasite ligands were involved in strong interactions with the RBC receptors and facilitate the deformation of the RBC seen during real time invasion visualisation. Here the authors use optical tweezers to measure the binding strength of merozoites to their host cells both using wild type but also mutants and with that managed to determine and actually measure that EBA175-GPA and RH4-CR1 in particular are significantly contributing to the binding strength and with that invasion success of merozoites. Whereas the majority of the data presented in this manuscript is not surprising, the methodology used here gives confirmation and quantitation and validation to previously accepted truths.

Reviewer #4: Kals et al apply optical tweezers to examine the contribution of different malaria merozoite surface exposed antigens to binding of the erythrocyte. The data in this study provides highly relevant and useful information to the field on what proteins actually contribute to binding and what inhibitors may actually act against it. This breaks the fields previous understanding which was largely based on indirect measures and speculation based on protein localisation. The technique, and therefore the study, has its limitations, but the authors provide a significant level of experimental detail to explore and mitigate these limitations as best as possible. The manuscript is mostly well written and the data completed to a high standard.

**Part II – Major Issues: Key Experiments Required for Acceptance**

Reviewer #1: The study is well conducted and I don't have any experimental suggestions but the written part of the paper should be improved by condensing the text (suggestions below)

Reviewer #2: The work in the paper is very comprehensive and performed to a high standard and so no major additional work is required.

Reviewer #3: Overall, this manuscript is an important scientific contribution to the field of invasion biology of Plasmodium parasites with the authors making use of their unique ability to measure merozoite attachment strength using their optical tweezer setup.

Although no additional data has to be provided to support the major conclusions in this manuscript some of the authors interpretations especially around the function of MSP1 are unsubstantiated and unless the authors are willing to conduct further experiments should be removed.

1. Lines 249-250: Giemsa-stained smears or SybrGreen-labelled FACS based parasitemia counting of cDELPfMSP1 cannot determine whether there is an invasion defect with these parasites. Given the documented egress defect of this mutant cell line, the lack of ring stages post egress, cannot be contributed to an invasion defect but an accumulative growth defect.

=> I was disappointed as the optical tweezer technology should allow the authors to manipulate individual cDELPfMSP1 merozoites from the stringy egress net and test whether they are, despite their egress defect, still able to invade red blood cells, but this experiment has not been done here. This would allow the speculation of whether MSP1 plays a role in egress AND invasion, or ONLY in egress to be settled.

2. lines 588-589 in Discussion state that as ‘cDELPfMSP1did not show a significant decrease in attachment or detachment force this is providing evidence for a role of MSP1 in egress’. Although detachment strength in this study was correlated with invasion success, like RH5 and AMA1 come after the strong binding stages of invasion, the weak strength interaction of MSP1 might facilitate the subsequent engagement of strong interactors such as EBA175. Knowing that MSP1 function is involved in egress does not rule out in addition a function in invasion. As stated above, the authors have a unique tool to decipher egress from invasion phenotype of cDELPfMSP1parasites. Please use it.

Reviewer #4: (No Response)

**Part III – Minor Issues: Editorial and Data Presentation Modifications**

Reviewer #1: 1 Title: Optical tweezers don’t per se reveal anything, please rephrase. It’s either their use or a particular type of use. Not sure you need to mention the technique in the title.

21 please delete ‘falciparum’ as there are different species causing malaria or rephrase. It might be worth to mention that the different parasite species do have different invasion ligands

29 not sure what the authors mean with ‘phenotypes’, maybe processes?

29-37 the abstract could in my view be written less defensively

47 ‘whilst using’ – maybe better: by combining it with… or by probing a set of…

57 ‘merozoite egress from the liver’ and ‘<60 sec’– merozoites are shed within merosomes and rupture in the lung, I don’t know of any study that quantified how quickly this happens, a minute would be a good estimate but no need to make a statement here that might be wrong.

73 citation 11 might be replaced with the more recent work showing that the parasites go in with the wider end. At least this should be added (PMID: 34819379).

97 I suggest to rephrase this sentence. ‘very low’ is not quantitative and some of the assays are rather high throughput in comparison to other assays. Maybe better to state that video microscopy is passive observation. This then allows a more elegant bridging to the optical tweezer assays. The following paragraph would benefit from not focusing solely on red bloo cell invasion but on the use of laser tweezers in general and then focus on the specific use case

99 Again, it’s not about quantitative, it’s about being able to manipulate

140 s15 video, should that not be video1?

157, 563 use ‘s’ instead of ‘sec’

165 please always only use two digits, e.g. 24 instead of 24.2

230-355 to enhance readability, the text could be compressed by not repeating details from the introduction and move technical issues to the materials and methods. Also for following pages of the results, it reads in parts more like a PhD thesis than like a paper.

In my view figure 4 and most of the associated text could go to supplement as it adds little as minor changes in mRNA might not at all be reflected on the protein level. It distracts from a beautiful biophysical study.

539 effectiency ?

541 However,… I am lost in this sentence, please rephrase

Is there anything known about attachment strength and frequency from other invasive parasite stage such as ookinetes or sporozoites or from T. gondii? If yes, please put your study into perspective. For those parasites optical tweezers have also been used.

A number of references are cited twice and should be merged

Reviewer #2: 1. It appears that the citations in the text to the figures do not always match up indicating the text and figures maybe from different drafts of the paper. These were FigsS3, S5 and S6, Figs 2 and 5. The paper needs to be very carefully checked before resubmission.

2. I am surprised heparin treatment was not assayed using the optical tweezers as merozoites still weakly attach to RBCs and herparin thought to interfere with MSP1 binding.

3. Why were EB175 knockout parasites not tested? EBA175 is known to bind GYPA and this interaction was blocked with anti-GYPA IgG but what if EBA175 binds to other receptors or the antibody does not completely prevent EBA175 from accessing GYPA?

4. The difference between the effects of R1 peptide which blocks AMA1 and the AMA1 knockout merozoites is noted. R1 increased the strength of merozoite binding and AMA1 knock out did not have an effect. Are the authors aware of https://doi.org/10.1007/s00018-023-04712-z where AMA1 was suggested to have a second receptor apart from RON2? Could the second interaction have kept merozoites engaged with the RBCs increasing the detachment force?

5. Line 138. Check ‘egressed so viable merozoite’

6. Line 156-157. Multiple names for seconds being used ie, sec and s.

7. Line 165. I think I know what “attachment frequency per egress” means but please clearly define.

8. Line 175. “p=0.0000” Really?

9. Fig S1B. Y-axis should be “Detachment Force”

10. Line 219. “cells that lead to attachment.” Attachment to what?

11. Line 317. Should be “anti-EBA140”?

12. Fig 2 legend. Text for “d and e” is missing.

13. Line 378 “0.952 when including them.” Should be excluding?

14. Line 511. Fig 4 should Fig 5?

15. Line 572. Could RBC age contribute to the variability observed?

16. Line 598. “The binding of several of the PfEBA175, PfEBA140…..” Do you mean ligands or knockout parasites?

17. Line 678 NaOH

18. Line 1097. Fig S3. End of legend discusses NF54 detachment data that is not in the figure.

19. Line 1113. WR99210

20. Line 1145. “..a little clumpy..” Use more accurate description that matches Table.

Reviewer #3: 1. In Lines 190-192 Fig S3 the authors point out that merozoite polarity does not significantly affect detachment score. Do the authors find this not surprising? Especially given that the strongest detachment score is measured with apically secreted ligands such as EBA175? Could the authors elaborate a bit more on this?

2. It would be helpful if the authors had quantitated their gene excision/invasion efficiencies (lines 247-248). Given antibodies have been used by the authors to assess whether excision has occurred (IFAs), the excision rate of the DiCre-mediated experiment could have easily been determined.

Although the overall data in this manuscript is important, the manuscript in its current form is full of errors/inaccuracies and needs a significant and careful re-assessment by the authors.

• At least in the pdf provided to the reviewers the readability and clarity of figures needs improvement. Firstly, the median is often not visible, the significance comparisons are hard to make out, the y-axes have no clear lines to determine where different % start and it would be helpful here to include not only lines at intervals of 10, but also at half of that.

• In other Figures such as Fig 3 it would be helpful to have the main data figure central and move sub-panels a) and b) to the supplementary as these are not data and only distract. The summary figure 6 however is very helpful and informative for the reader.

By lines/figures:

• Lines 172-174: PEMR should read as dividing the ring parasitemia by the parasitemia the day before and not the other way around.

• FigS4 WR99210, not WT9921

• Lines 1144-1146- S5Fig legend: grammar/spelling

• Lines 1149 S5Fig which statistical testing was used here? “a test”

• Lines 254 – 256: referring to cDELPfMSP1attachment strength measuring but does not refer to a figure where these data should be verifiable.

• Lines 271 -272: “efficient gene excision with no band, indicating the presence of unedited parasites in the rapamycin-treated samples”? Obviously, this is not what is meant here, rather the opposite.

• Lines 272-273: There is no panels f or g in S5 Figure. And where do the 1.43% invasion efficiency data come from? In S5 Fig that invasion efficiency is listed as 0% - 4.3%.

• Fig S6 c) were the authors not surprised that with 40uM of R1 peptide the parasites still achieved ~20% of their normal invasion rate? Are these 20% real, living parasites?

• Line 296: “Neuraminidase treatment reduced invasion by nearly half..” where is this data displayed?

• Line 298: Fig 2d, not 2e

• Line 309: “anti-GYPA invasion was 55.5%...” in which figure is that data shown?

• Line 310 Fig2d

• Line 312 Fig 2d

• Line 321 Fig 2d

As no Fig 2e or f exists

• Line 333 insert Figure number you show this anti-Basigin data

• Line 337 Fig 2d

• Line 347 Fig 2d

• Line 354 S5 Fig e does not exist

• Line 374 is not clear- please spell out “there was a weak correlation…” (between which two variables?).

• S7 Fig b) is unclear. According to the diagram I would expect C,D only to amplify with wt locus primers; whereas A and B should not. Panel b only confuses. Instead list one primer pair number in the graphic in a) and then go to PCR data images.

• Fig 4: has no panel f although that is listed in legend

• S10 Fig (line 1205) b-c should read a-b as no panel c exist

• Line 511: Fig5, not Fig4

• Fig 6 correct spelling mistakes

Reviewer #4: -In general, the manuscript is well written, but there are quite a lot of small errors, some of which I put further down. The manuscript would benefit from a good round of editing.

-An EBA175 knock-out parasite is briefly mentioned several times in the manuscript, but does not actually seem to be used for the experiments. Given the interpretation around the importance of EBA175 for binding through GLYA experiments, I find this surprising and it would obviously improve the manuscript if this data was available and included. This would greatly strengthen the support for EBA175 being important for binding. However, it is not included and so I find this distracting and I am not sure why the data is there. If the data was done then it would be best to include it. If it is to be used in another paper then it would be best to state this and remove any reference to the EBA175 knock-out where it is not needed. At this stage, as written, it does not seem to be needed.

-The measurement of Frequency of Attachment is a useful measure as merozoites can sometimes attach to multiple cells or multiple times to a cell before invading. In this study, these measures were done on wildtype parasites. Could the authors speculate on whether surface exposed merozoite antigens could have a role in this multiple attachment potential and whether this could be explored using knock-out lines? Although most knock-outs used in this study show no change in erythrocyte binding, I wonder whether the ability to reattach might also be important in a flow environment.

-Why would GAP45 impact on initial attachment? Typically, crude measures of disrupting the invasion motor by CytoD lead to disruption of entry. The paper cited did not report any change in merozoite binding to erythrocytes (whether they looked at this is not mentioned), only in deformation. Biologically, I don’t really see why the glideosome would be linked to attachment frequency. Can the authors discuss this observation and possible mechanism some more.

-For the CR1 data, it might be worth exploring the limitations of other inhibitors (soluble CR1, antibodies) given the need for neuraminidase treatment to see a growth defect. This would provide a rationale for why these additional steps may not have been taken in this study to explore Rh4 and CR1.

-As I understand the data, the neuraminidase treatments lead to a 15-fold loss in attachment frequency, but only a 2 fold loss in invasion. This does not seem possible. Can the authors speculate why? Could this indicate a difficulty in using the frequency measure or how that is assessed?

-The current dataset suggests that EBA15 and Rh4 are linked to attachment, but not the other EBAs and Rhs. Yet these proteins are considered to have redundancy in function. Do the authors consider that redundancy does not include the function in attachment? Or is this an indication of the limits of the knock-outs where it is known that some lines naturally do not rely on certain EBAs and Rhs and so knock-outs tend to have minimal impact. I think it would be worth clarifying this beyond an interest in double knock-outs etc.

-Figure 2A: spelling of merozoite.

-Any reason why both 6 and 8 cycles were used for PEMR in the study?

-Figure 6: spelling of Merozoite (x2)

-Units: By convention, there should be a gap between the number and the SI unit. Throughout the text some measures have a space and some do not.

-Below I provide some passages where I suggest the wording could be improved or words removed/added (this is not exhaustive so some more review would be suggested).

Line 138: used to manipulate a newly egressed so viable merozoite and position

Line 187: Line starting ‘Merozoites are polar cells..’ and the subsequent paragraph. I found the concepts explored here difficult to follow as written. If is clearer in Fig S3. I suggest making this passage clearer.

Line 296: There ‘was’ also significantly reduced attachment.

Line 297: for neuraminidase-treated erythrocytes (1.4 ± 0.5%), ‘which was’ significantly different to NF54 (rank-sum p=0.0001, Fig 2e);

Line 319: of levels observed ‘in’ the absence of the antibody.

Line 424: For all genes, this insertion was made within the predicted erythrocyte binding domain for all genes other than PfRH4 (repeated text).

Line 429: receptors were known. (where?).

Line 539: effectiency

Line 543: as if invasion is blocked after attachment has occurred, then it is expected that the invasion rate would be reduced.

Line 544: completion. (completeness??)

Line 607: face of increase shear. (increasing??)

Line 647: role of PfMSP1 not in. (is not??)

Line 654: the NF54 background, we used, the interactions of PfEBA175 and PfRH4 654 appear. (too many commas?)

Line 720: erythrocytes the coverslip. (to the??)

Line 732: the erythrocyte length (Lo) of each erythrocyte. (probably don’t need the first erythrocyte). It is also worth looking at the wording at this section. The sentence just mentioned implies that measures were made for each erythrocyte. The final sentence is not clear what it means: If two erythrocytes were stretched, then

only the detachment force was recorded.

Line 735: F = kΔL,. (comma?)

A couple of times: FSCA vs FSC-A. A couple of your flow cytometry methods are quite similar. Could be worth considering having a single separate method?

Line 827: after the invasion widow. (window?)

Line 852: The selectable was. (wording needs changing???)

Line 880: while the R1 peptide (VFAEFLPLFSKFGSRMHILK) (GenScript at 98% purity). (wording needs changing???)

There is a repeat in text between acknowledgements and financial disclosure. May not be necessary?

Line 1070: detachment force of merozoite-erythrocyte detachment force. (repeated text could be reworded?)

Line 1088: are summaries. (summarised??)

S3Fig 1: (c-d) Show……. Then (d) shows. (This happens a few times. Capitals or not for all?

Line 1145: The table table. (repeated?)

Line 1150: (c) cΔPfAMA1. (d) cΔPfGAP45. (These descriptions are a bit brief. Worth expanding?).

-Any reason why the downstream (truncated) region of PfRH4 was not amplified when checking knock-out?

PLOS authors have the option to publish the peer review history of their article (what does this mean?). If published, this will include your full peer review and any attached files.

Reviewer #1: No

Reviewer #2: No

Reviewer #3: No

Reviewer #4: No
---

## [Decision Letter · Decision Letter 1]

16 Jul 2024

Dear Ms Kals,

Thank you very much for submitting your manuscript "Application of optical tweezer technology reveals that PfEBA and PfRH ligands, not PfMSP1, play a central role in Plasmodium-falciparum merozoite-erythrocyte attachment" for consideration at PLOS Pathogens. As with all papers reviewed by the journal, your manuscript was reviewed by members of the editorial board and by several independent reviewers. 

The reviewers were very positive about the revised version of your manuscript and the text was much improved. As you can see from their comments, there is just one small issue that needs to be resolved. Based on the reviews, we are likely to accept this manuscript for publication, providing that you modify the manuscript according to the last remaining reviewer recommendation.

Sincerely,

Tobias Spielmann

Academic Editor

PLOS Pathogens

Jeffrey Dvorin

Section Editor

PLOS Pathogens

Michael Malim

Editor-in-Chief

PLOS Pathogens

orcid.org/0000-0002-7699-2064

Reviewer Comments (if any, and for reference):

Reviewer's Responses to Questions

**Part I - Summary**

Reviewer #1: As stated in my initial review, this is an interesting study that needed essentially a lot of textual improvement, which the authors now provided.

Reviewer #2: (No Response)

Reviewer #3: (No Response)

Reviewer #4: No further comments.

**Part II – Major Issues: Key Experiments Required for Acceptance**

Reviewer #1: none

Reviewer #2: (No Response)

Reviewer #3: (No Response)

Reviewer #4: No further comments.

**Part III – Minor Issues: Editorial and Data Presentation Modifications**

Reviewer #1: there are some small typos that can be easily corrected in proof stage

Reviewer #2: The authors have satisfactorily address most issues of concern except for Fig S13c.

Fig S13c is mentioned 2x on page 18. There is no S13c so change to a or b.

Reviewer #3: (No Response)

Reviewer #4: No further comments.

PLOS authors have the option to publish the peer review history of their article (what does this mean?). If published, this will include your full peer review and any attached files.

Reviewer #1: No

Reviewer #2: No

Reviewer #3: **Yes: **Dr Ellen Knuepfer

Reviewer #4: No

Figure Files:

Data Requirements:

Reproducibility:

References:

---

## [Editor Report · Decision Letter 2]

5 Aug 2024

Dear Ms Kals,

We are pleased to inform you that your manuscript 'Application of optical tweezer technology reveals that PfEBA and PfRH ligands, not PfMSP1, play a central role in Plasmodium-falciparum merozoite-erythrocyte attachment' has been provisionally accepted for publication in PLOS Pathogens.

Best regards,

Tobias

Tobias Spielmann

Academic Editor

PLOS Pathogens

Jeffrey Dvorin

Section Editor

PLOS Pathogens

Michael Malim

Editor-in-Chief

PLOS Pathogens

orcid.org/0000-0002-7699-2064
---

## [Editor Report · Acceptance letter]

17 Sep 2024

Dear Ms Kals,

We are delighted to inform you that your manuscript, "Application of optical tweezer technology reveals that PfEBA and PfRH ligands, not PfMSP1, play a central role in Plasmodium-falciparum merozoite-erythrocyte attachment," has been formally accepted for publication in PLOS Pathogens.

Best regards,

Michael Malim

Editor-in-Chief

PLOS Pathogens

orcid.org/0000-0002-7699-2064